# Structure by Architecture: Structured Representations without Regularization

**Felix Leeb** [*]**, Giulia Lanzillotta, Yashas Annadani, Michel Besserve, Stefan Bauer, & Bernhard Schölkopf**
Max Planck Institute for Intelligent Systems, Tübingen, Germany

## Abstract

We study the problem of self-supervised structured representation learning using autoencoders for downstream tasks such as generative modeling. Unlike most methods which rely on matching an arbitrary, relatively unstructured, prior distribution for sampling, we propose a sampling technique that relies solely on the independence of latent variables, thereby avoiding the trade-off between reconstruction quality and generative performance typically observed in VAEs. We design a novel autoencoder architecture capable of learning a structured representation without the need for aggressive regularization. Our *structural decoders* learn a hierarchy of latent variables, thereby ordering the information without any additional regularization or supervision. We demonstrate how these models learn a representation that improves results in a variety of downstream tasks including generation, disentanglement, and extrapolation using several challenging and natural image datasets.

## 1 Introduction

Deep learning has achieved strong results on a plethora of challenging tasks. However, performing well on a carefully tuned or synthetic datasets is usually insufficient to transfer to real-world problems (Tan et al., 2018; Zhuang et al., 2019) and directly collecting real labeled data is often prohibitively expensive. This has led to a particular interest in learning representations without supervision and designing inductive biases to achieve useful structure in the representation to help with downstream tasks (Bengio et al., 2013; Tschannen et al., 2018; Bengio et al., 2017; Tschannen et al., 2018; Radhakrishnan et al., 2018).

Autoencoders (Vincent et al., 2008) have become the de-facto standard for representation learning largely due the simple self-supervised training objective paired with the enormous flexibility in the model structure. In particular, variational autoencoders (VAE) (Kingma & Welling, 2013a) have become a popular framework which readily lends itself to structuring representations by augmenting the objective or complex model architectures to achieve desirable properties such as matching a simple prior distribution for generative modeling (Kingma & Welling, 2013a; Vahdat & Kautz, 2020; Preechakul et al., 2022), compression (Yang et al., 2022), or disentangling the underlying factors of variation (Locatello et al., 2018; Khrulkov et al., 2021; Shu et al., 2019; Chen et al., 2020; Nie et al., 2020; Mathieu et al., 2019; Kwon & Ye, 2021; Shen et al., 2020).

However, VAEs exhibit several well-studied practical limitations such as posterior collapse (Lucas et al., 2019b; Hoffman & Johnson, 2016; Vahdat & Kautz, 2020), holes in the representation (Li et al., 2021; Stühmer et al., 2020), and blurry generated samples (Kumar & Poole, 2020; Higgins et al., 2017; Burgess et al., 2018; Vahdat & Kautz, 2020). Consequently, we seek a representation learning method which serves as a drop-in replacement for VAEs while improving performance on common tasks such as generation and disentanglement. One promising direction of the problem is causal modeling (Pearl, 2009; Peters et al., 2017; Louizos et al., 2017; Mitrovic et al., 2020; Shen et al., 2020) which formalizes potential non-trivial structure of the generative process using structural causal models (see appendix A.1 for further discussion). This powerful and exciting framework serves as inspiration for our main contributions:

---

[*]Email: fleeb@tuebingen.mpg.de

- We propose an architecture called the *Structural Autoencoder* (SAE), where the *structural decoder* infuses latent information one variable at a time to induce an intuitive ordering of information in the representation without supervision.

- We provide a sampling method, called *hybrid sampling* which relies only on independence between latent variables, rather than imposing a prior latent distribution thereby enabling more expressive representations.

- We investigate the generalization capabilities of the encoder and decoder separately to better motivate the SAE architecture and to assess how the learned representation of an autoencoder can be adapted to novel factors of variation.

## 1.1 RELATED WORK

The most popular autoencoder based method is the Variational Autoencoder (VAE) (Kingma & Welling, 2013a), which has inspired the whole family of disentanglement focused methods including the $\beta$-VAE (Higgins et al., 2017), FactorVAE (Kim & Mnih, 2018) or the $\beta$-TCVAE (Chen et al., 2018). These methods aim to match the latent distribution to a known prior distribution by regularizing the reconstruction training objective (Locatello et al., 2020; Zhou et al., 2020). Although this structure is convenient for generative modeling and even tends to disentangle the latent space to some extent, it comes at the cost of somewhat blurry images due to posterior collapse and holes in the latent space.

To mitigate the double-edged nature of VAEs (Makhzani & Frey, 2017; Lin et al., 2022), less aggressive regularization techniques have been proposed such as the Wasserstein Autoencoder (WAE), which focuses on the aggregate posterior (Tolstikhin et al., 2018). Unfortunately, WAEs generally fail to produce a particularly meaningful or disentangled latent space (Rubenstein et al., 2018), unless some supervision is available (Han et al., 2021).

Meanwhile, a complementary approach to carefully adjusting the training objective is designing a model architecture beyond the conventional feed-forward style to induce a hierarchy in the representation. For example, the variational ladder autoencoder (VLAE) (Zhao et al., 2017) separates the latent space into separate chunks each of which is processed at different levels of the encoder and decoder (called "rungs"). However, due to the regularization, VLAEs suffer from the same trade-offs as conventional VAEs. Further architectural improvements such as FiLM (Perez et al., 2018) or Ada-In layers (Karras et al., 2019) readily learn more complex relationships resulting in more expressive models.

## 2 METHODS

Given (high-dimensional) $X = (X_1, \ldots, X_D)$, the **encoder** learns to encode the observations into a low-dimensional representation $U = (U_1, \ldots, U_d)$ ($d \ll D$), and the **decoder** models the generative process that produced $X$ from the latent variables to produce reconstruction $\hat{X}$. To structure the representation, without loss of generality, we may imagine the true generative process to follow some graphical structure between the (unknown) true factors of variation.

If the graphical structure of the true generative process was known, the topology of the decoder could be shaped accordingly (Kipf & Welling, 2016; Zhang et al., 2019; Lin et al., 2022; Yang et al., 2021). However, in the fully unsupervised setting, we seek a sufficiently general structure on top of the uninterpretable hidden layers to make the representation more structured and interpretable.

## 2.1 STRUCTURAL DECODERS

*Structural decoders* integrate information into the generative process one latent variable at a time, resulting in the following structure:

$$S_i := f_i(S_{i-1}, U_i), \quad (i = 1, \ldots, d), \tag{1}$$

where $S_i$ is a feature map ($S_0$ are static sample-independent trainable parameters) that depends on their noise $U_i$, their parent $S_{i-1}$ and indirectly any ancestors $S_j$ for $j < i - 1$ through $S_{i-1}$.

We choose $f_i = \text{ConvBlock}^n \circ \text{StrTfm}$ where $\text{ConvBlock}^n$ is some number of convolution blocks (where $n$ depends on the dataset and can freely be adjusted) and $\text{StrTfm}$ is a Structural-Transform layer. Given the latent variable $U_i$, the $i$th Str-Tfm layer applies a pixelwise affine transform to the given feature map $S_{i-1}$ as seen in figure 1. The Str-Tfm layers are based on FiLM layers (Perez et al., 2018) and the related Ada-In layers (Karras et al., 2019) (except, crucially, only a single latent variable is used for each Str-Tfm layer), which have been shown to infuse latent information into feature maps well compared to naive alternatives such as concatenation.

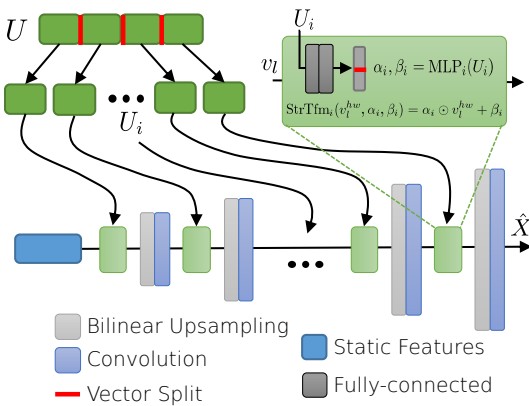

Figure 1: The structural decoder reconstructs (or generates) a sample from a latent vector $U$ by first splitting $U$ into $d$ variables each of which infuses latent information with an affine transforms of the pixel $v_l^{hw}$ in image feature map $S_i$ produced by a Str-Tfm layer (green box where $\alpha_i$ and $\beta_i$ are the affine parameters that are extracted from the latent variable $U_i$ by network $\text{MLP}_i$).

There are three main features of this architecture: (1) since each latent variable is infused at a different layer of the decoder, the decoder capacity for $U_i$ is larger than for $U_j$ when $i < j$. This biases high-level non-linear information towards the earlier (and thereby deeper) latent variables, while the later variables can only capture relatively low-level linear features with respect to the observation space. (2) The independently initialized hidden layers of each Str-Tfm layer bias the Str-Tfm layers to be independent of one another (similar to the "lambdas" in Bello (2021)). (3) The affine transform for each Str-Tfm layer is initialized to identity, so each latent variable has to independently learn to affect the generative process selecting for informative latent variables. This has a similar effect as the regularization term of the VAE, except instead of penalizing informative latent variables directly and thereby conflicting with the reconstruction objective and giving rise to posterior collapse, it is only a weak architectural bias. Overall, this architectural asymmetry between latent variables encourages statistical independence and induces a relatively intuitive hierarchical structure of the latent space.

Note that, without supervision, there is a priori no guarantee that the learned structure will recover the true one (Locatello et al., 2018). Instead, the model is only optimized to reproduce the same observational distribution as the true generative process. Disentanglement methods are commonly evaluated with synthetic datasets where the factors of variation are independent by design (Träuble et al., 2021). For methods penalizing correlations, this implicitly biases the latent variables to align with the true factors. However, the view that disentanglement can be reduced to statistical independence has been contested (Träuble et al., 2021; Schölkopf et al., 2021), which is where a more causal approach may provide additional insight. We treat disentanglement as measured by common metrics not as the objective of our model, but merely as a tool to better understand the structure of the representation. Independence of latent variables implies that interventions on them do not violate the learned generative process (i.e., the decoder), thereby enabling hybrid sampling as an alternative to variational regularization techniques for generative modeling.

## 2.2 Hybrid Sampling and Objective

For generative modeling, it is necessary to sample novel latent vectors that are transformed into (synthetic) observations using the decoder. Usually, this is done by regularizing the training objective so the posterior matches some simple prior (e.g. the standard normal). However, in practice, regularization techniques can fail to adequately match the prior and actually exacerbate the information bottleneck, leading to blurry samples from holes in the learned latent distribution and unused latent dimensions due to posterior collapse (Dai & Wipf, 2019; Stühmer et al., 2020; Lucas et al., 2019a; Hoffman & Johnson, 2016). Instead of trying to match some prior distribution in the latent space, we suggest an alternative sampling method that eliminates the need for any regularization, which in turn also makes it easier to augment the objective with auxiliary loss terms in settings where additional supervision is available. Consequently, the training objective for our models here is only the reconstruction error (specifically, the cross entropy loss).

Inspired by Besserve et al. (2018), we refer to this sampling method as *hybrid sampling*. The goal is to draw new latent samples $\tilde{U}$ from a distribution having independent variables with marginals matching those observed in the training set, leading to $p(\tilde{U}) \approx \prod_i p(U_i)$. In practice, we first store a set of $N$ (= 128 in our case) latent vectors $\{U^{(j)}\}_{j=1}^N$, selected uniformly at random from the training set. We then generate latent samples from $\tilde{U}$ by choosing independently the value for each variable $\tilde{U}_i$, which is sampled uniformly at random from the corresponding set of values of this variable in the stored vectors $\{U_i^{(j)}\}_{j=1}^N$.

This allows the model to generate a diversity of samples well beyond the size of the training set ($N^d$ distinct latent vectors), spanning the Cartesian product of the supports of the (marginal) distribution of each $U_i$ on the training data, which includes the support of the joint latent distribution of $U$ observed during training. Note that hybrid sampling is directly applicable to any learned representation as it does not affect training at all, however the fidelity of generated samples will diminish if there are strong correlations between latent dimensions. Consequently, the goal is to achieve maximal independence between latent variables without compromising on the fidelity of the decoder (i.e., reconstruction error), which for SAEs is done through the architecture alone. This also aligns with the approach of existing unsupervised disentanglement methods, which suggests the sampling method may serve as a promising alternative to the VAE-based prior-based sampling, beyond just SAEs.

## 3 EXPERIMENTS

We train the proposed methods and baselines on two smaller disentanglement image datasets (where $D = 64 \times 64 \times 3$ and the true factors are independent): 3D-Shapes (Burgess & Kim, 2018) and the three variants ("toy", "sim", and "real") of the MPI3D Disentanglement dataset (Gondal et al., 2019), as well as two larger more realistic datasets (where $D = 128 \times 128 \times 3$): Celeb-A (Liu et al., 2015) and the Robot Finger Dataset (RFD) (Dittadi et al., 2020).

The models are trained with a standard 70-10-20 (train-val-test) split of the datasets where the training objective uses the cross entropy loss (as well as the method specific regularization terms for the baselines). We evaluate the quality of the reconstructions based on the reconstruction loss and the Fréchet Inception Distance (FID) (Heusel et al., 2017) as in Williams et al. (2020) on the test set. The FID is able to capture higher level visual features and can be used to directly compare the reconstructed and generated sample quality, while the binary cross entropy is a purely pixelwise comparison.

Next we compare the performance of the hybrid sampling method to the prior based sampling. Unlike the prior based sampling, which only makes sense for the models that use regularization , the hybrid sampling method can be applied to any latent variable model. Finally we take a closer look at the representations to understand how the model architecture affects the induced structure and disentanglement.

### 3.1 MODELS

Since the datasets involve images, all models use the same CNN backbone for both the encoder and decoder with the same number of convolution blocks (see the appendix for details). For the smaller datasets, the encoder and decoder have 12 convolution blocks, the latent space has 12 dimensions in total, and the models are trained for 100k iterations, while for CelebA and RFD each the encoder/decoder has 16 convolution blocks each with twice as many filters, the latent space is 32 dimensional in total, and the models are trained for 200k iterations.

We compare four kinds of autoencoder architectures. The first type is our Structural Autoencoders (SAE) which use a conventional deterministic encoder and a structural decoder with the latent space split evenly into 2, 3, 4, 6, or 12 variables for the smaller datasets and 16 variables for the larger ones corresponding to the labels SAE-2, SAE-3, SAE-4, SAE-6, SAE-12 and SAE-16 respectively. Consequently, the "structural" architecture has an architectural asymmetry between latent variables in the decoder, but not in the encoder. The simplest "baseline" architecture uses the traditional "hourglass" architecture, in addition to a variety of different regularization methods: (1) unregularized autoencoders ("AE"), (2) Wasserstein-autoencoders ("WAE") (Tolstikhin et al., 2018), (3) VAE (Kingma & Welling, 2013b), (4) $\beta$VAE (Higgins et al., 2017), (5) FactorVAE ("FVAE") (Kim

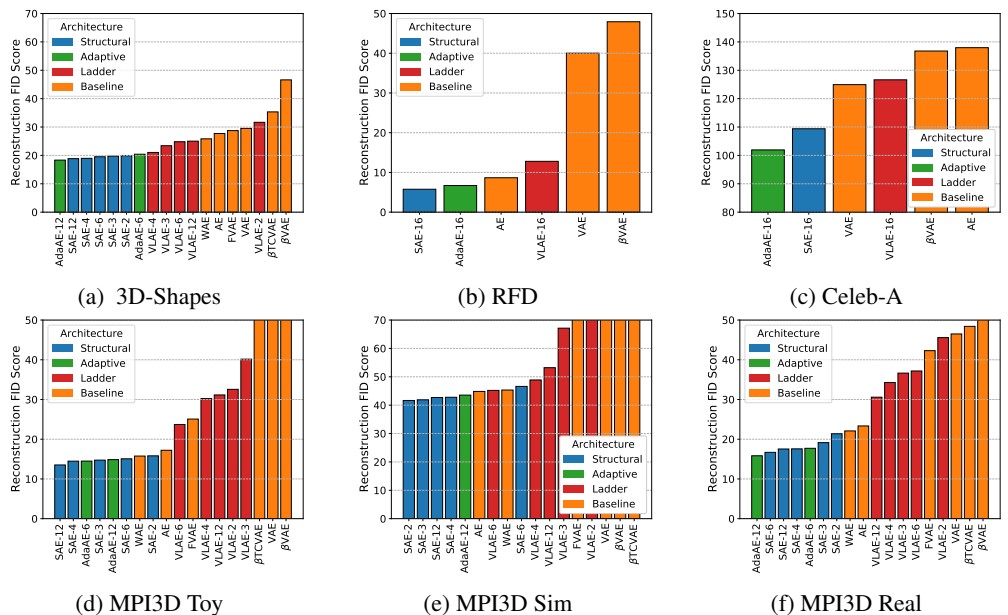

Figure 2: Reconstruction quality for all models and datasets (lower is better). "Baseline" models correspond to traditional "hourglass" CNN architectures, while the "Structural" models use our novel architectures to further structure the learned representation.

& Mnih, 2018), and (6) $\beta$-TCVAE (using the experimentally determined best hyperparameters, see the appendix A.2.2). The next baseline architecture is called "AdaAE" (and referred to as "Adaptive"). It is identical to the SAE models except that all latent variables are passed to each of the Str-Tfm layers, so the architecture has connections between the latent variables and intermediate layers, but without the architectural asymmetry. For the purposes of hybrid sampling, each latent dimension of these less structured baselines is treated as a separate latent variable. The final type of architecture we investigate is the variational ladder autoencoder (Zhao et al., 2017) which also learns a hierarchical representation, but unlike the structural autoencoders, VLAEs also use the variational regularization and use an encoder architecture that roughly mirrors the decoder, consequently both the encoder and decoder break the architectural symmetry between latent variables. Just like for the SAEs, we include variants of the VLAEs with 2, 3, 4, 6, 12, and 16 rungs.

## 3.2 EXTRAPOLATION

Since VLAEs structure both the encoder and decoder, while SAEs only structure the decoder, we aim to better characterize the relative behaviors of the encoder and decoder. Specifically, this analysis aims to understand to what extent the encoder or decoder is the "weaker link" in regards to integrating new information into the representation.

First, both the encoder and decoder are trained jointly on a subset of 3D-Shapes where only three distinct shapes exist (instead of four, as the ball is missing) for 80k iterations. Then either the encoder only, the decoder only, or both are trained for another 20k iterations on the full 3D-Shapes training dataset. The reconstruction error for samples not seen during any part of training is compared for each of the variants and each of the architectures to identify how well the encoder extrapolates compared to the decoder, and to assay implications for designing novel autoencoder architectures.

## 4 RESULTS

In terms of reconstruction quality, the structural autoencoder (SAE) architecture consistently outperforms the baseline methods (figure 2). As expected, unregularized methods like the SAE, AdaAE, and AE tend to have significantly better reconstruction quality. However, also noteworthy is that the structured architectures SAE and VLAE show improved results compared to their unstructured

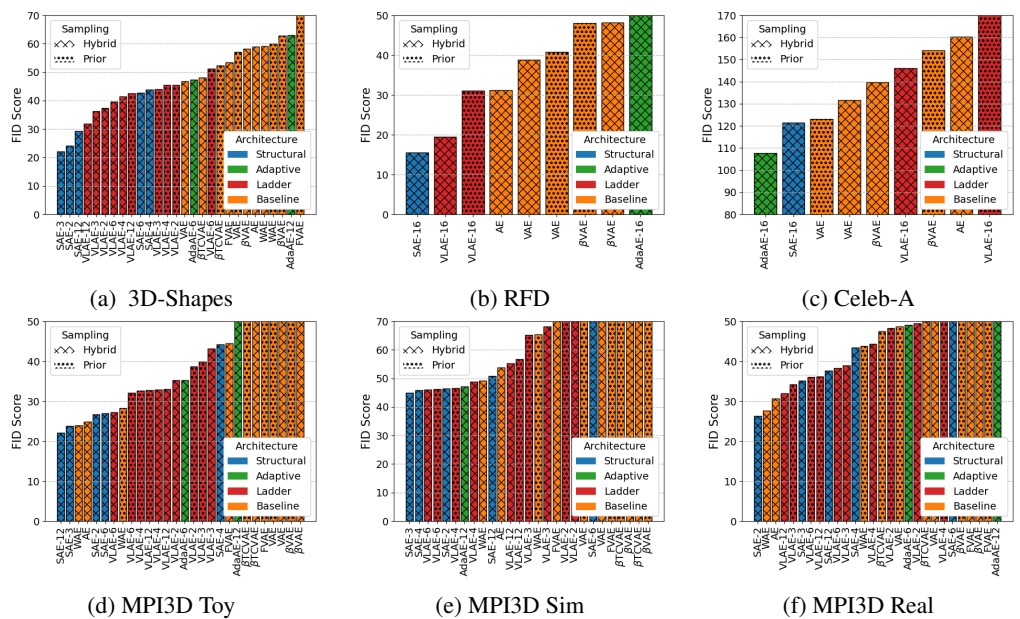

Figure 3: Quality of the generated samples using different models and sampling methods (lower is better). Note that our SAE models perform well without having to regularize the latent space towards a prior. In fact, even with the conventional "hourglass" architecture (in orange), the hybrid sampling method generates relatively high quality samples, often outperforming the more principled prior-based sampling.

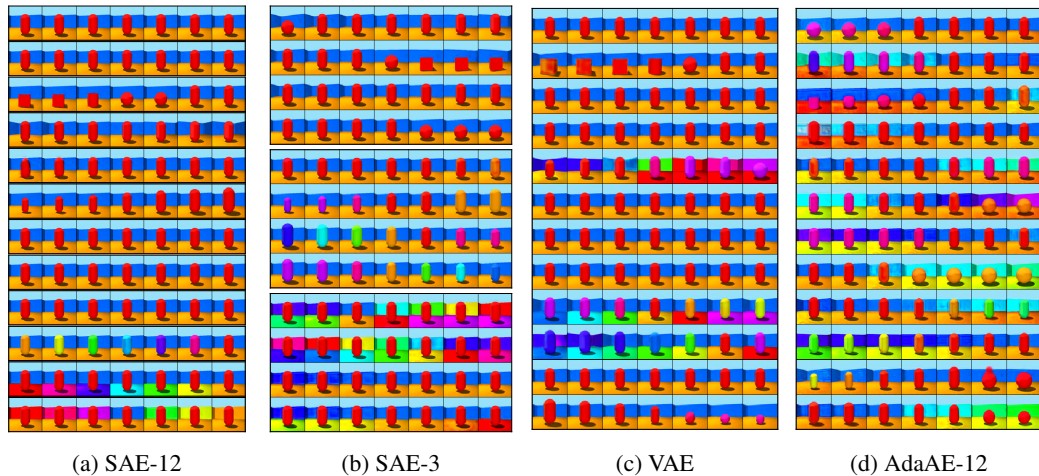

Figure 4: Latent traversals of several models trained on 3D-Shapes, in their original order. Note the ordering of the information in the structural decoder models (SAE-12 and SAE-3) where higher level, nonlinear features (like shape and orientation) are encoded in the first few dimensions, which feed into Str-Tfm layers deeper in the network. Somewhat surprisingly, the SAE-12 even learns to compact the representation by ignoring superfluous latent variables (e.g. first row) resembling the effect of posterior collapse in VAEs (see A.3.1).

counterparts. It should be noted that there is some controversy regarding the reliability of FID with synthetic datasets (Razavi et al., 2019; Barratt & Sharma, 2018). However, we consistently find strong agreement between the reconstruction FID and the pixelwise metrics (see figure 8 in the appendix for further discussion).

| Model | DCI | MIG | IRS | Mod | Exp |
|---|---|---|---|---|---|
| SAE-12 | **0.974** | 0.537 | **0.830** | **0.967** | **1.000** |
| SAE-6 | 0.865 | 0.225 | 0.735 | 0.966 | 0.999 |
| SAE-4 | 0.740 | 0.209 | 0.654 | 0.945 | 0.999 |
| VLAE-12 | 0.832 | **0.553** | 0.751 | 0.914 | 0.977 |
| VLAE-6 | 0.785 | 0.326 | 0.689 | 0.929 | 0.963 |
| VLAE-4 | 0.690 | 0.282 | 0.544 | 0.900 | 0.926 |
| $\beta$TCVAE | 0.410 | 0.237 | 0.603 | 0.865 | 0.923 |
| FVAE | 0.330 | 0.123 | 0.725 | 0.907 | 0.955 |
| $\beta$VAE | 0.235 | 0.127 | 0.593 | 0.879 | 0.799 |
| VAE | 0.314 | 0.138 | 0.607 | 0.892 | 0.872 |
| WAE | 0.211 | 0.050 | 0.621 | 0.946 | 0.906 |
| AE | 0.307 | 0.092 | 0.638 | 0.926 | 0.943 |
| AdaAE-12 | 0.299 | 0.062 | 0.503 | 0.876 | 0.999 |

Figure 5: Disentanglement scores for 3D-Shapes. DCI denotes the DCI disentanglement score (Eastwood & Williams, 2018), MIG is the Mutual Information Gap (Chen et al., 2018), IRS is the Interventional Robustness Score (Suter et al., 2018), and Mod/Exp refers to the Modularity/Explicitness scores respectively (Ridgeway & Mozer, 2018) (for all these metrics higher is better). The figure on the right shows how the scores vary across five models with different random seeds marked with a cross (lines indicate the resulting mean and standard deviation). Both hierarchical methods, SAE-12 and the VLAE-12, outperform all other baselines, and in particular the SAE performs well, despite the lack of regularization.

Comparing the SAE models to the AdaAE architecture, we see that there can be a slight penalty in reconstruction quality incurred from separately processing the latent variables. However, this is more than made up for in the quality of the generated samples (shown in figure 3), where the SAE models perform significantly better than the baselines. Even the regularized models such as VLAE, FVAEs, and VAEs consistently generate higher quality samples using the hybrid sampling than when sampling from the prior they were trained to match (also in figure 6b). Surprisingly, the AdaAE architecture actually outperforms all other models on CelebA using hybrid sampling. This may be explained by the severity of the information bottleneck experienced when embedding CelebA into only 32 dimensions. Consequently, the higher fidelity decoder from the intermediate connections of the latent vector exceed the performance penalty incurred by the hybrid sampling when disregarding the correlations between latent variables.

In general, if we consider the latent distribution, then sampling from the approximated factorized prior can introduce at least two types of errors: (1) errors due to not taking into account statistical dependences among latent variables, and (2) errors due to sampling from "holes" in the latent distribution if the prior does not match it everywhere. Whenever (2) is the dominating source of error, hybrid sampling is preferred. Furthermore, learning independent latent variables aligns with the aim towards disentangled representations, while minimizing the divergence between the posterior and prior results in a compromise that does not necessarily promote disentanglement.

## 4.1 HIERARCHICAL STRUCTURE

To get a rough idea of how the representations learned using the structural decoders differ from more conventional architectures, figure 4 shows the one dimensional latent traversals (i.e., each row corresponds to the decoder outputs when incrementally increasing the corresponding latent dimension at a time from the min to the max value observed). The traversals illustrate the hierarchical structure in the representation learned by the SAE models: the information encoded in the first few latent variables can be more nonlinear with respect to the output (pixel) space, as the decoder has more layers to process that information, while the more linear information must be embedded in the last few variables. This results in a reliable ordering of "high-level" information (such as object shape or camera orientation) first, followed by the "low-level" information (such as color). This means the structural decoder architecture biases the representation to separate and order the information necessary for reconstruction (and generation) in a meaningful way, and thereby ordering, and potentially even fully disentangling, the underlying factors of variation better.

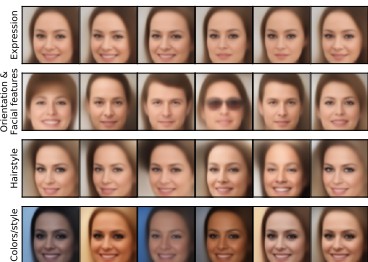

(a) Here we use hybridized chunks of latent vector to show the diffuse disentanglement achieved by the hierarchical structure of the SAE-16 architecture when combined with the flexible hybrid sampling. For each row the corresponding quarter of latent dimensions (8/32) are hybridized (see section 2.2) while the remaining dimensions are fixed. This shows how the SAE architecture is able to order partially disentangled factors of variation from high-level (more nonlinear, like facial expressions and features) to low-level (such as color/lighting) without any additional regularization or supervision.

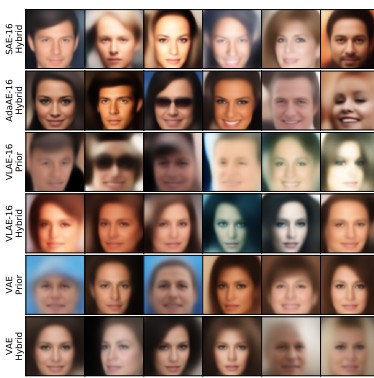

(b) Samples generated using hybrid and prior-based sampling using several models trained on CelebA. Note that the hybrid sampling tends to produce relatively high quality samples both for our proposed SAE and AdaAE architectures as well as baselines.

Figure 6: CelebA controllable generation and sampling comparison

Figure 5 evaluates how disentangled the representations are, using common metrics. The table shows the disentanglement scores of the same models discussed above, while the plot on the right sheds light on quality of the representations varies for five different random seeds (used to initialize the network parameters). Most noteworthy is that the SAE-12 model consistently achieves very high disentanglement scores. This shows, empirically, that the SAE architecture promotes independence between latent variables (especially SAE-12). We may explain this as a consequence of splitting up the latent dimensions so that each variable has a unique parameterization in the decoder, making different latent variables less likely to transform the feature map of the decoder in the same way.

Unsurprisingly, when there are multiple latent dimensions per variable (like in SAE-6, SAE-4, etc.), the dimensions within a variable are entangled similarly to the baselines like AE or adaptive baselines. Since all of these disentanglment metrics are computed on a dimension-by-dimension basis, the resulting scores are systematically underestimated. Qualitatively, these SAE models still achieve the same ordering of causal mechanisms, as can be seen from figure 4b.

For a real world demonstration of how well SAE models are able to order information in the latent space, figure 6a shows generated CelebA samples when varying only a quarter of the latent variables at a time, with the labels on the left describing roughly the semantic semantic information contained. Note that the inductive biases are not strong enough to fully disentangle the factors of variation into individual latent dimensions. However, the hierarchical structure learns a diffuse kind of disentanglement where information pertaining to higher-level features tend to be encoded in the first few dimensions while lower level factors of variation show up towards the last few dimensions.

SAE models achieve this structured disentanglement using the Str-Tfm layers as opposed to the standard FiLM/Ada-In layers. Each Str-Tfm layer only has access to one of the latent variables which is not directly seen by any other part of the decoder. In contrast, the Ada-In layers used by the AdaAE allows information from anywhere in the latent vector to leak into any part of the decoder. Consequently, the AdaAE does not disentangle the representation at all, as seen in the table of figure 5 (although it achieves impressive results for reconstruction nonetheless, see figure 2).

## 4.2 EXTRAPOLATION

Results of the experiment described in section 3.2 are shown on figure 7. Perhaps unsurprisingly, none of the models were particularly adept at zero-shot extrapolation to observations that were not in the initial training data, as is consistent with Schott et al. (2021). Comparing the first two columns

on the right, without any update, the reconstructed images filter out the novel information in the sample (in this case, ball shape), and instead reconstruct a similar sample seen during training (a cylinder).

If only the encoder is updated on some observations with the additional shape while the decoder is frozen, then the reconstruction performance increases somewhat, but some deformations and artifacts become visible in the reconstruction. This suggests the frozen decoder struggles to adequately extrapolate, even when the encoder extends the representation to include the ball.

In contrast, when only the decoder is updated, the reconstructions qualitatively look much more similar to the original observations. Although the encoder can be expected to generally filter out any information it has not been trained to encode into the latent space due to the bottleneck, the representation may still extrapolate somewhat provided the decoder can reconstruct any novel features. This underscores the importance of focusing on carefully shaping the decoder, as embodied by the SAEs, because the decoder is not able to extrapolate as well as the encoder.

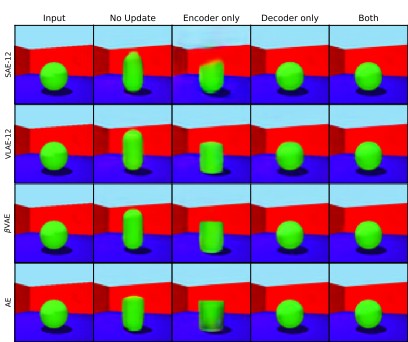

| Model | Neither | Encoder | Decoder | Both |
|---|---|---|---|---|
| SAE-12 | 13.21 | 7.7 | 0.42 | **0.34** |
| SAE-6 | 13.35 | 7.99 | 0.52 | 0.36 |
| VLAE-12 | 18.37 | 7.69 | 1.55 | 0.62 |
| VAE | 12.97 | 8.78 | 0.44 | 0.46 |
| $\beta$VAE | 15.49 | 8.31 | 1.35 | 0.52 |
| AE | 11.81 | **7.31** | 0.38 | 0.35 |
| WAE | **11.68** | 7.87 | **0.37** | 0.35 |

Figure 7: Average reconstruction error (MSE x1000) on novel observations when training either the encoder, decoder, both, or neither on the initially left-out ball shape (see section 3.2). Example input and reconstructed images are shown above. All models perform significantly better when updating the decoder than the encoder, and reach a reconstruction quality that is almost indistinguishable from the model when updating both the encoder and decoder. Furthermore, the SAE-12 generally outperforms all variational baselines, suggesting the aggressive regularization of VAEs makes updating the representation more difficult.

## 5 CONCLUSION

While VAEs provide a principled approach for generative modeling with autoencoders, in practice, the regularization tends to suppress the dependence of the posterior on the observations by minimizing its discrepancy to the prior (Hoffman & Johnson, 2016; Tolstikhin et al., 2018), resulting in a trade-off between reconstruction quality and matching the prior. While regularization may lead to some disentanglement, this tends to have negative effects on sample quality, and due to the identifiability problem, disentanglement cannot be guaranteed (Locatello et al., 2018).

This motivated us to develop an alternative paradigm that improves sample fidelity while still achieving comparable disentanglement. The key is to simplify the training objective and exploit the architectural flexibility of deep models instead. We show that despite the identifiability problem, architectural biases can make the representation more interpretable by focusing on a diffuse but intuitive ordering of semantic information, rather than the full dimension-by-dimension disentanglement. Meanwhile, our hybrid sampling technique leverages the independence between latent variables achieved by both VAE-based and architecture based, rather than requiring the learned aggregate posterior to match some unstructured prior. This unifies the goals of achieving a samplable representation and one with independent latent variables, thereby addressing some of the commonly accepted weaknesses of existing VAE-based methods.

While it is encouraging how far one can get with suitable architectural biases, future work should further investigate links between the learned generative process and the true causal mechanisms to make the representation more interpretable and potentially enhance generalization. For instance, we could explicitly encourage the realism of hybrid samples, or the sparsity of latent factor changes across domain shifts based on the Sparse Mechanism Shift Hypothesis (Schölkopf et al., 2021).

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

# A  APPENDIX

## A.1  LINKS TO CAUSAL REPRESENTATION LEARNING

The structural decoders can be interpreted as a structural causal model, thereby linking the causal modeling community to representation learning. A *structural causal model (SCM)* represents the relationship between random variables $S_i$ using a directed acyclic graph (DAG) whose edges indicate direct causation and *structural assignments* of the form

$$S_i := f_i(\mathbf{PA}_i, U_i), \quad (i = 1, \ldots, D), \tag{2}$$

encoding the dependence of variable $S_i$ on its parents $\mathbf{PA}_i$ in the graph and on an "unexplained" noise variable $U_i$ (Pearl, 2009). The noises $U_1, \ldots, U_D$ are assumed to be jointly independent. Any joint distribution of the $S_i$ can be expressed as an SCM using suitable $f_i$ and $U_i$. Note, that although structural decoders define the relatively simple sequential graphical structure $\mathbf{PA}_i = S_{i-1}$, as long as the capacity of $S_i$ is sufficiently large, all DAGs can be represented by this structure. Furthermore, the SCM contains additional information regarding how statistical dependencies between the $S_i$ are *generated* by mechanisms (2), such that changes due to *interventions* can be modelled as well (e.g., by setting some $U_i$ to constants). From this perspective, hybrid sampling can be interpreted as interventions on the learned generative process. Although thus far we focus on the relatively naive approach of jointly intervening on all latent variables for unconditional generative modeling, as shown in figure 6a.

We construct the causal ordering as follows: we evaluate the $f_i$ of a root node $i$, i.e., $f_i$ depends only on $U_i$. In the step, we evaluate any node $j$ which depends only on its $U_j$ and possibly other variables that have already been computed. We iterate until there are no nodes left. This terminates (since the graph is acyclic) and yields a unique $f(U)$, but the order $\pi(i)$ in which the $f_i$ get evaluated need not be unique. It is referred to as a *causal* or *topological* ordering (Peters et al., 2017), satisfying $\pi(i) < \pi(j)$ whenever $j$ is a descendant of $i$. This embeds the **SCM** into the network starting from the bottleneck $U = (U_1, \ldots, U_D)$ with the $U_i$ feeding into subsequent computation layers according to a causal ordering. This structure reflects the fact that the root node(s) in the DAG only depend on "their" noise variables, while later ones depend on their noise and those of their parents, and so on.

Recall that for a *causally sufficient* system, the set of noises $U_1, \ldots, U_n$ are assumed to be jointly independent. If, in contrast, only a subset of the causal variables are modelled, then the noises will in the generic case be dependent. We would expect that the architectural bias implemented by the structural decoder, however, may still be a sensible one.

In this work, we evaluate the proposed architectural biases exclusively on image datasets as these serve as a particularly intuitive and well-studied setting for disentanglement research due to the high-dimensional observations with relatively few underlying factors of variation. Nevertheless, the structural decoder architecture can directly be applied to data without spatial structure (since the Str-Tfm layers are applied pixel-wise anyway), and the hybrid sampling method is agnostic to the model architecture and training (as long as the learned latent variables are mutually independent).

## A.2  TRAINING PROCEDURE

### A.2.1  ARCHITECTURE DETAILS

As described in the main paper, the basic convolutional backbone of all models is the same. For the smaller datasets, 3D-Shapes and the MPI3D datasets (where observations are 64x64 pixels), the encoder and decoder each have 12 convolutional blocks. Each block has a convolutional layer with 64 channels and a kernel size of 3x3 and stride of 1 (unless otherwise specified), followed by a group normalization layer and then a MISH nonlinearity (Misra, 2019). In the encoder, the features are downsampled using a 2x2 Max Pooling layer right after the convolution every third layer starting with the first one and the first convolution layer uses a kernel size of 5x5. In the decoder, every third convolution layer is immediately preceded by a 2x2 bilinear upsampling. For our structured modules (SAE and AdaAE), the specified number of Str-Tfm layers are placed evenly in between the convolution blocks. For SAE models, the latent space is always split evenly between Str-Tfm layers, and each layer uses a three hidden layer network to process the latent space segment into the scale and bias vectors which are then applied to all pixels individually of the features. The output layer

biases of each Str-Tfm network are initialized to 0 and the output corresponding to the $\alpha$ scaling is interpreted as logits (just as is done for the standard deviation parameters of the posterior of VAEs), such that $\alpha_i \approx 1$ and $\beta_i \approx 0$ for all Str-Tfm layers at the beginning of training. For the VLAE models, the inference and generative ladder rungs each also have a three hidden layers to process the features into and out of the separate latent space segments respectively.

While the latent space was always 12 dimensional for 3D-Shapes and MPI3D, for Celeb-A we use a 32 dimensional latent space. For Celeb-A, we also expand the 12 block backbone to 16 blocks and double the filters per convolution layer to 128. The exact sizes and connectivity of the models can be seen in the configuration files of a the attached code, but overall, each of the 3D-Shapes and MPI3D models have approximately 1-1.2M trainable parameters, while for CelebA the models have 6-7M parameters. Note that the VLAE models always have the most parameters due to the hierarchical structure in the encoder that mirrors the decoder, the SAEs and AdaAEs have roughly 5% fewer parameters, and almost the same as the conventional hourglass architecture of the remaining baselines (the Str-Tfm layers contribute negligible additional parameters relative to the CNN backbone).

### A.2.2 TRAINING DETAILS

All models used the same training hyperparameters, which included using an Adam optimizer with a learning rate of 0.0005 and momentum parameters of $\beta_1 = 0.9$ and $\beta_2 = 0.999$. For the smaller datasets (3D-Shapes, MPI3D) the models were trained for 100k iterations and a batch size of 128, while for Celeb-A and RFD the models were trained for 200k iterations and a batch size of 32. The hyperparameters for the RFD dataset the same as for Celeb-A, except that the number of channels per convolution layer was doubled and the learning rate was decreased by a factor of 10.

The models are implemented using Pytorch (Paszke et al., 2019) and were trained on the in-house computing cluster using Nvidia V100 32GB GPUs, so that training a single model takes about 3-4 hours on the smaller datasets and 7-10 hours for CelebA.

For the $\beta$-VAEs and $\beta$-TCVAEs, $\beta \in \{2, 4, 6, 8, 16\}$, while $\gamma \in \{10, 20, 40, 80\}$ for the FVAE were tested on 3D-Shapes and MPI3D, and the model with the smallest loss on the validation set was used for subsequent analysis, which was $\beta = 2$ for the $\beta$-VAE and $\beta = 4$ for the $\beta$-TCVAE and $\gamma = 40$ for the FVAE. All other method-specific hyperparameters were kept the same in the corresponding papers.

### A.3 ADDITIONAL RESULTS

### A.3.1 ARCHITECTURALLY INDUCED INDEPENDENCE BETWEEN LATENT VARIABLES

A closer look at figure 4a shows traversing several of the latent variables of the SAE-12 has no effect on the resulting image (e.g. the first and second rows) (also observed in the other datasets as in figures 11, 13, 14, 15, and 18. This resembles the well-studied phenomenon in VAE-based models of posterior collapse (e.g. first row of figure 4c) where some latent variables are unused/uninformative to avoid incurring the regularization cost. However, for SAEs, since there is no regularization loss, instead we can understand this behavior as a consequence of the independence between latent variables due to the Str-Tfm layers.

Specifically, since at the beginning of training all Str-Tfm layers map roughly to identity, the decoder must learn to extract useful information from the latent variable through the Str-Tfm layer, otherwise the latent variable has no effect on the decoding process at all. This biases the decoder to only use those latent variables that are in fact informative, otherwise it is preferable for the decoding process to ignore the variable entirely to keep the affine transform of the Str-Tfm layer close to identity. Consequently, the use of Str-Tfm layers biases the decoder to treat each latent variable independently and thereby only use the variables that contain information useful for reconstruction not already contained in a different variable. Although this is obviously only a weak bias, it is evidently sufficient to learn a more compact representation without incurring as much of a cost to the sample fidelity as observed with posterior collapse of VAEs.

### A.3.2 3D-SHAPES

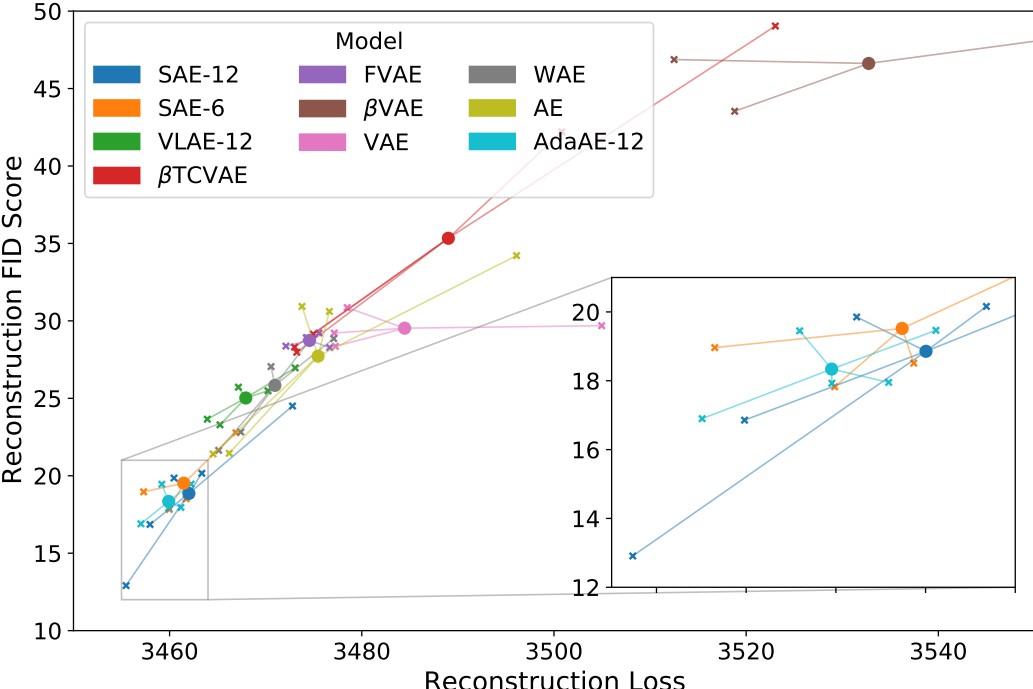

Figure 8: 3D-Shapes reconstruction quality comparison between models using the reconstruction loss (binary cross entropy) and the Fréchet Inception Distance (FID) (Heusel et al., 2017) between the original and reconstructed observations (lower is better for both). Each "x" is a model trained with a unique random seed using the architecture/regularization corresponding to the color. The performance of all the seeds are averaged and plotted as circles "o". Firstly, this plot shows how the reconstruction FID (y-axis) can complement the pixelwise comparison (x-axis) to quantify the quality of the reconstructed samples. Next, the multiple seeds help differentiate the performance of the regularized vs unregularized methods. These regimes separate the models that use the variational regularization loss from the models that only use a reconstruction loss (or a regularization on the aggregated posterior like the WAE). Lastly, the AdaAE-12 slightly out performs the SAE-12 and both of which significantly out perform less structured baselines, which suggests the architectural biases are conducive to high fidelity decoders.

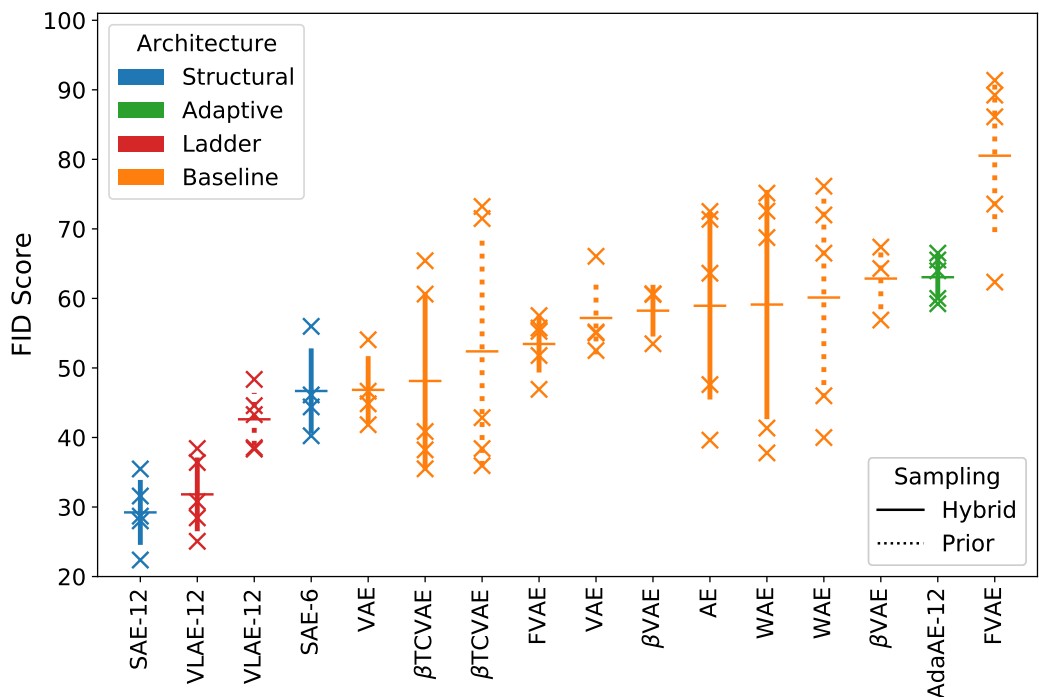

Figure 9: Comparison of the hybrid and prior-based sampling method for all different models. Each "x" corresponds to a unique random seed, the horizontal line corresponds to the mean performance and the vertical line signifies one standard deviation above and below the mean. (lower is best)

| Model | DCI | MIG | IRS | Mod | Exp |
|---|---|---|---|---|---|
| SAE-12 | **0.974** | 0.537 | **0.830** | **0.967** | **1.000** |
| SAE-6 | 0.865 | 0.225 | 0.735 | 0.966 | 0.999 |
| SAE-4 | 0.740 | 0.209 | 0.654 | 0.945 | 0.999 |
| VLAE-12 | 0.832 | **0.553** | 0.751 | 0.914 | 0.977 |
| VLAE-6 | 0.785 | 0.326 | 0.689 | 0.929 | 0.963 |
| VLAE-4 | 0.690 | 0.282 | 0.544 | 0.900 | 0.926 |
| $\beta$TCVAE | 0.410 | 0.237 | 0.603 | 0.865 | 0.923 |
| FVAE | 0.330 | 0.123 | 0.725 | 0.907 | 0.955 |
| $\beta$VAE | 0.235 | 0.127 | 0.593 | 0.879 | 0.799 |
| VAE | 0.314 | 0.138 | 0.607 | 0.892 | 0.872 |
| WAE | 0.211 | 0.050 | 0.621 | 0.946 | 0.906 |
| AE | 0.307 | 0.092 | 0.638 | 0.926 | 0.943 |
| AdaAE-12 | 0.299 | 0.062 | 0.503 | 0.876 | 0.999 |

Table 1: Disentanglement and Completeness scores for 3D-Shapes. The DCI-d metric corresponds to the DCI-disentanglement score and DCI-c to the completeness score (Eastwood & Williams, 2018), IRS is a similar disentanglement metric (Suter et al., 2018), the MIG is the Mutual Information Gap (Chen et al., 2018), and Mod/Exp refers to the Modularity/Explicitness scores respectively (Ridgeway & Mozer, 2018) (for all these metrics higher is better)

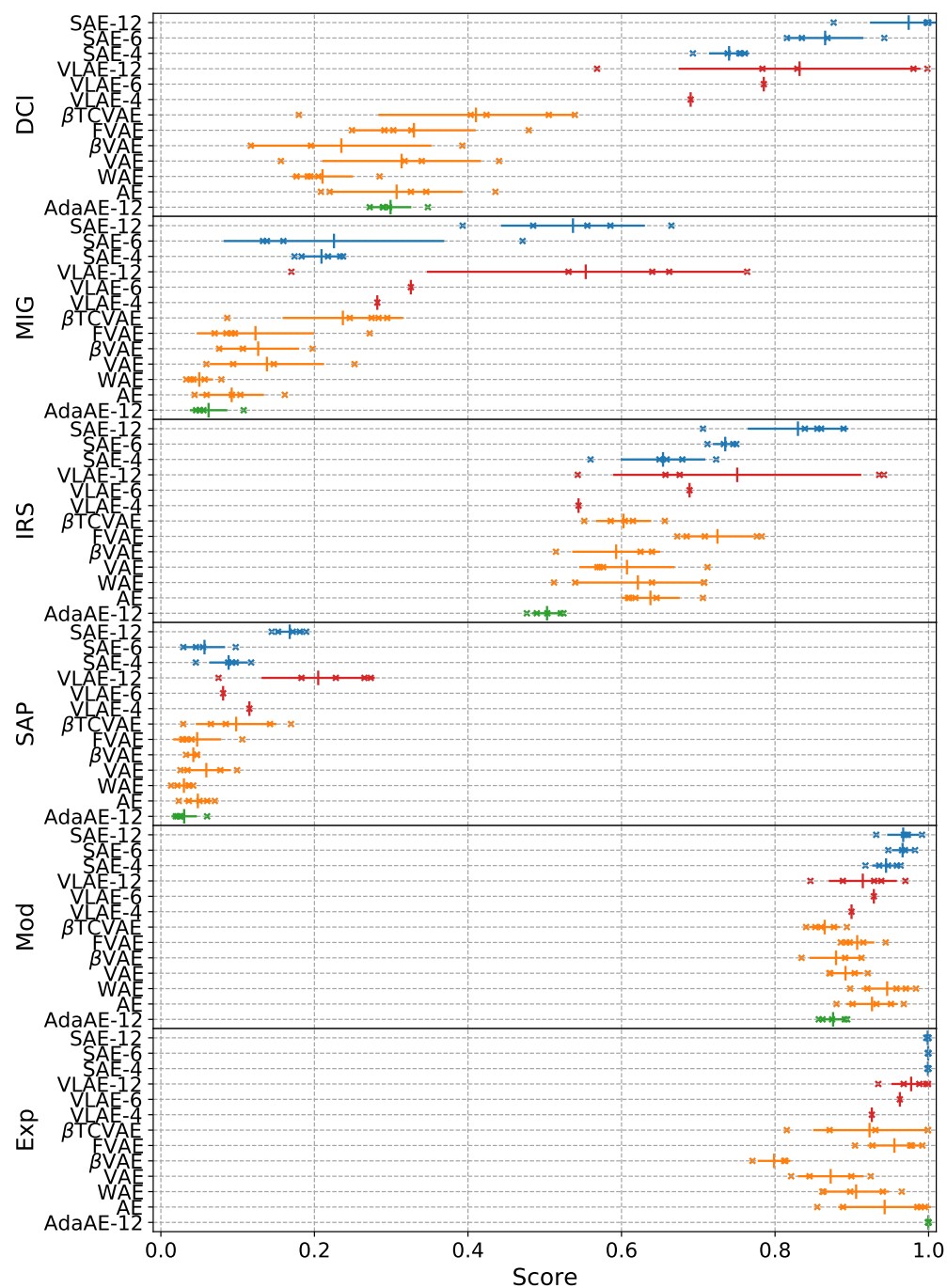

Figure 10: Several disentanglement metrics for all the models and each of the seeds. (for all these metrics higher is better)

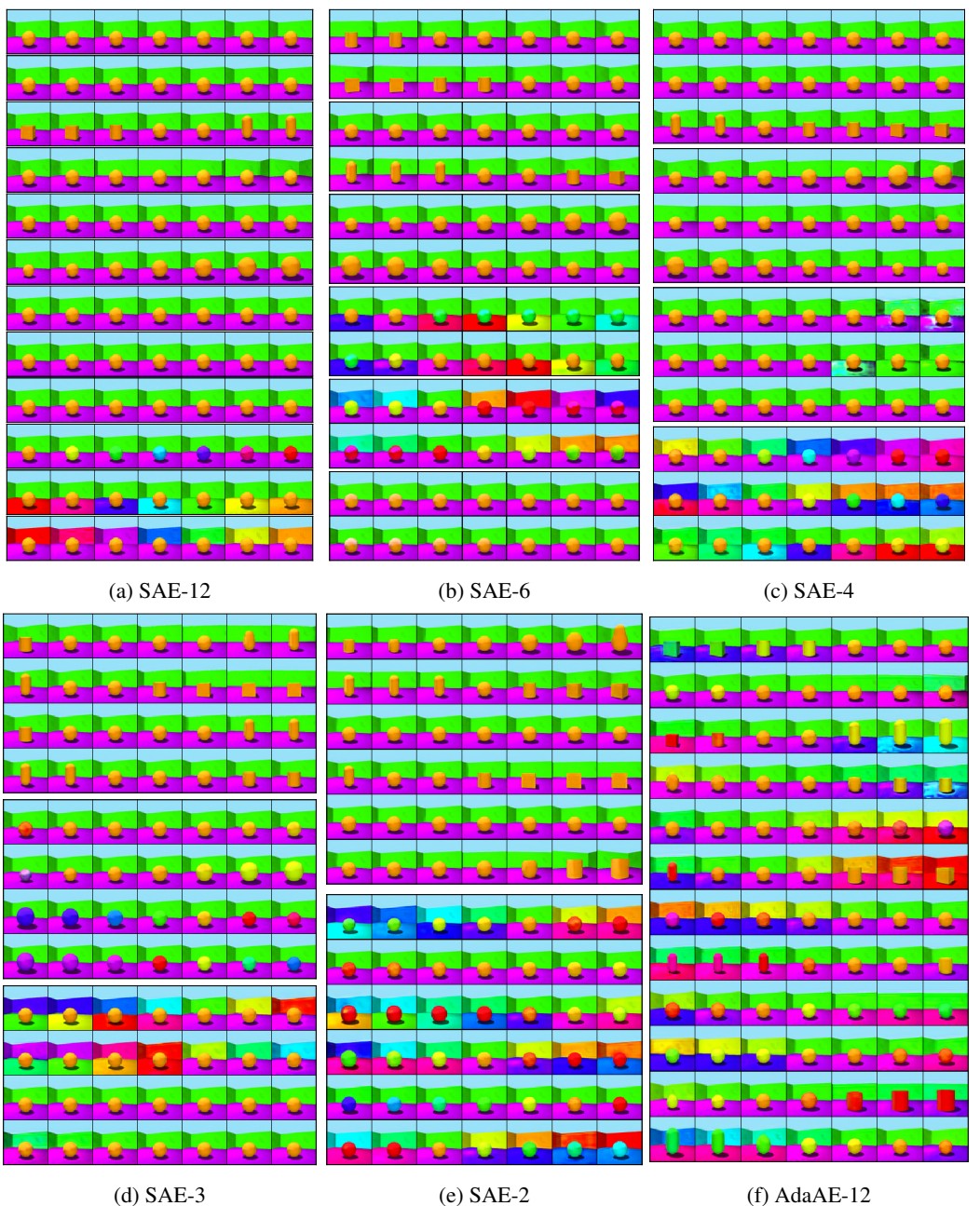

(a) SAE-12    (b) SAE-6    (c) SAE-4

(d) SAE-3    (e) SAE-2    (f) AdaAE-12

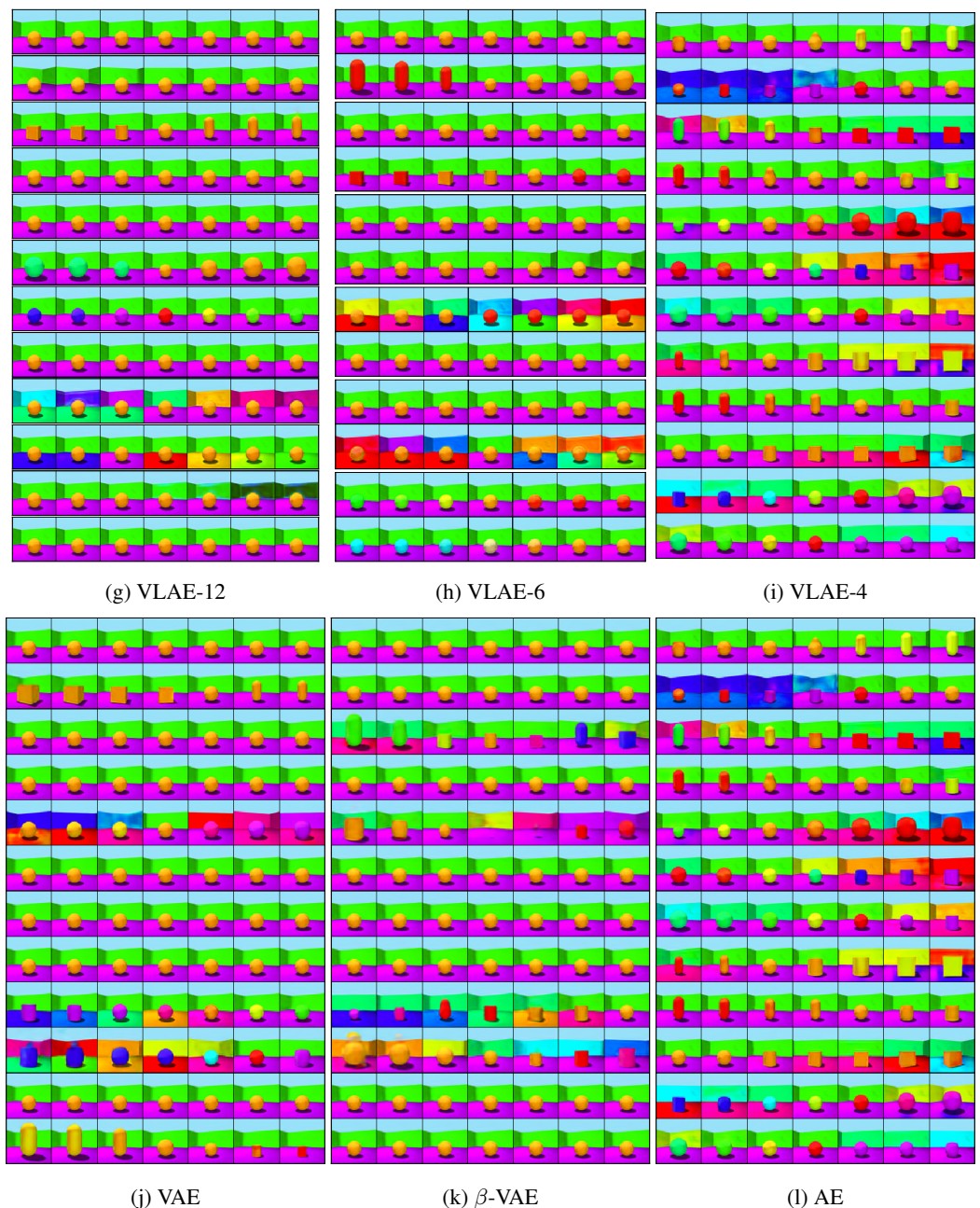

Figure 11: Latent Traversals of several models for 3D-Shapes. Each row shows the generated image when varying the corresponding latent dimension while fixing the rest of the latent vector. For the SAE and VLAE models, the groups of dimensions that are fed into the same Str-Tfm layer (or ladder rung) are grouped together. Note the disentangled segments achieved by the SAE models and the consistent ordering of factors of variation.

### A.3.3 EXTRAPOLATION

We present results on a variant of the exptrapolation experiment discussed in section 3.2. Instead of modifying the shape in the initial training dataset, we remove the two most extreme camera angles in either direction (removing 4/15 of the full dataset).

As seen from figure 12, in this setting the warping and generally lower fidelity experienced by only updating the encoder compared to updating the decoder is very apparent.

| Model | Neither | Encoder | Decoder | Both |
|---|---|---|---|---|
| SAE-12 | 4.88 | **2.65** | 0.77 | 0.43 |
| VLAE-12 | 6.91 | 3.83 | 1.61 | 0.77 |
| AdaAE-12 | 4.8 | 2.93 | **0.59** | **0.41** |
| AE | 4.98 | 2.94 | 0.62 | 0.45 |
| WAE | **4.79** | 3.07 | 0.69 | 0.44 |
| VAE | 5.17 | 3.09 | 1.01 | 0.51 |
| $\beta$VAE | 5.7 | 3.97 | 1.5 | 0.82 |

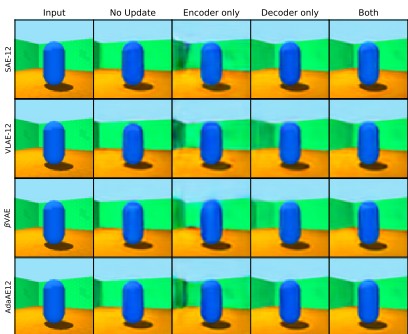

Figure 12: Same as figure 7, except for the camera angle setting. Although the results are generally very consistent, note how much better the SAE-12 model performs than the VLAE-12 in this setting.

### A.3.4 MPI3D-TOY

| Model | DCI | MIG | IRS | Mod | Exp |
|---|---|---|---|---|---|
| SAE-12 | **0.642** | **0.487** | 0.570 | 0.938 | **0.946** |
| SAE-6 | 0.454 | 0.094 | 0.553 | 0.918 | 0.911 |
| VLAE-12 | 0.414 | 0.323 | 0.667 | 0.909 | 0.842 |
| $\beta$TCVAE | 0.091 | 0.007 | 0.605 | 0.858 | 0.678 |
| FVAE | 0.108 | 0.029 | 0.680 | 0.876 | 0.743 |
| $\beta$VAE | 0.046 | 0.004 | **0.987** | **0.998** | 0.621 |
| VAE | 0.093 | 0.078 | 0.621 | 0.861 | 0.659 |
| WAE | 0.203 | 0.028 | 0.633 | 0.904 | 0.859 |
| AE | 0.186 | 0.043 | 0.632 | 0.911 | 0.844 |
| AdaAE-12 | 0.208 | 0.080 | 0.546 | 0.919 | 0.931 |

Table 2: Disentanglement and Completeness scores for MPI3D-Toy. (for all these metrics higher is better)

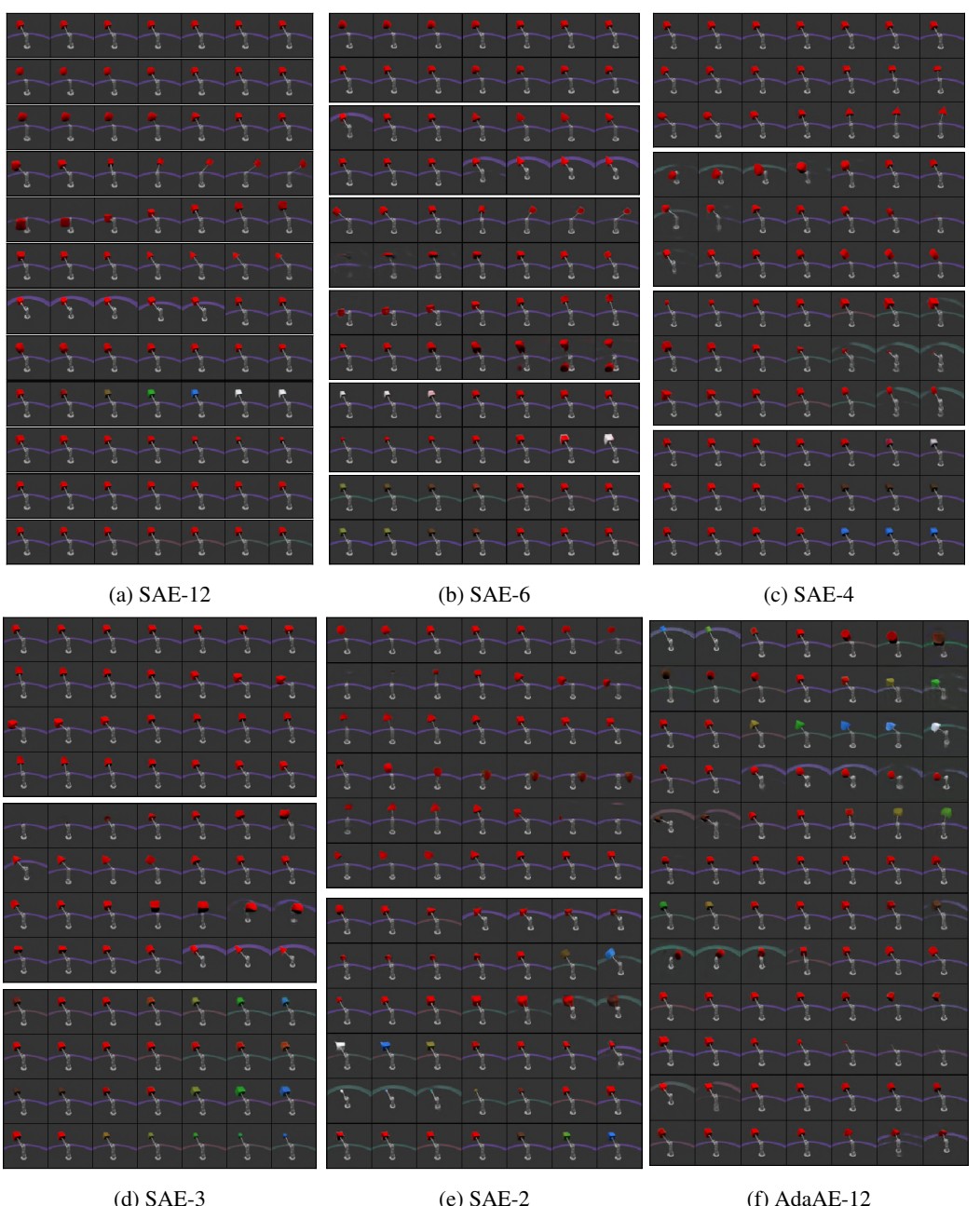

(a) SAE-12       (b) SAE-6       (c) SAE-4

(d) SAE-3       (e) SAE-2       (f) AdaAE-12

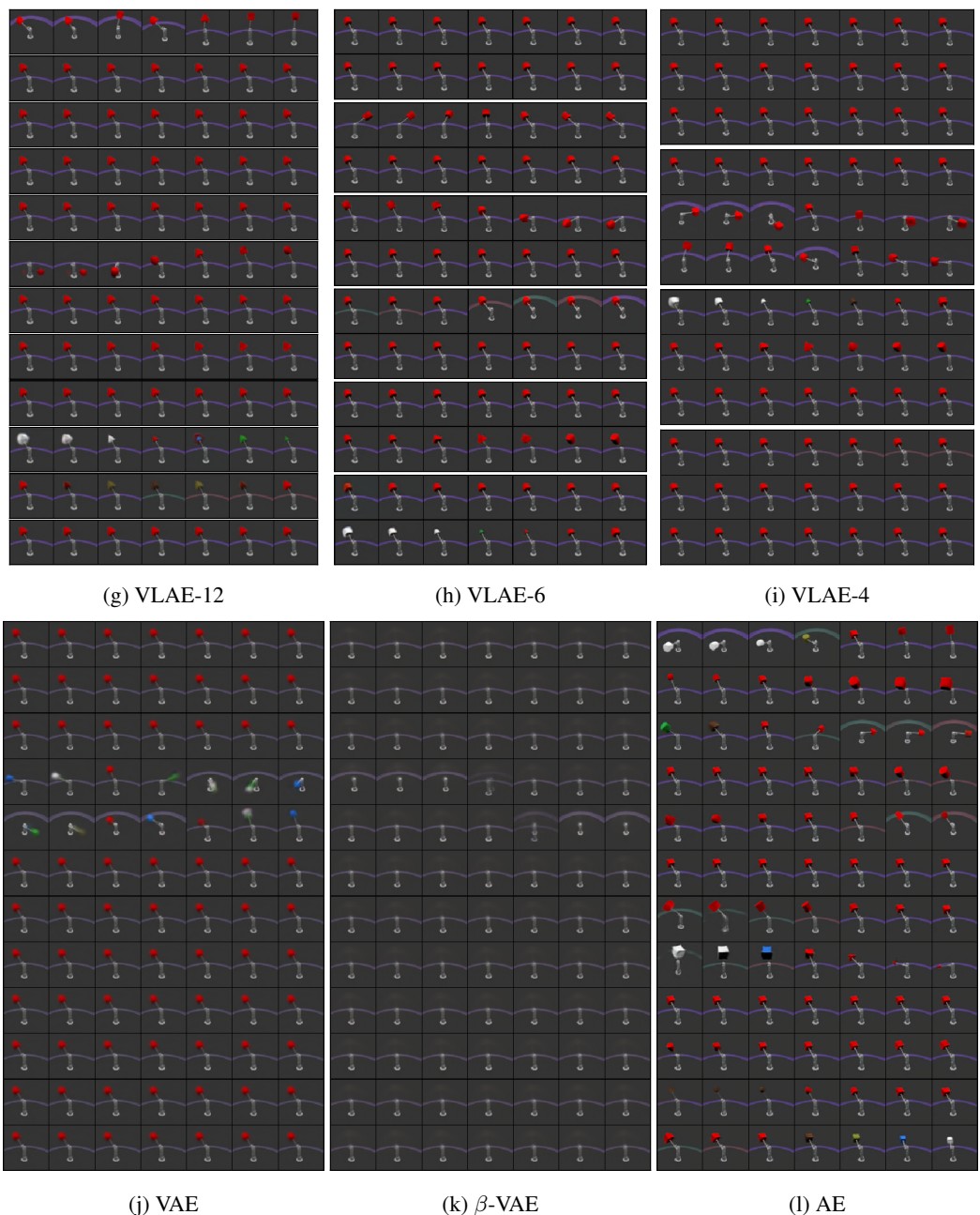

Figure 13: Latent Traversals of several models for MPI3D-Toy. Each row shows the generated image when varying the corresponding latent dimension while fixing the rest of the latent vector. For the SAE and VLAE models, the groups of dimensions that are fed into the same Str-Tfm layer (or ladder rung) are grouped together. Note the disentangled segments achieved by the SAE models and the consistent ordering of factors of variation.

### A.3.5    MPI3D-Sim

| Model | DCI | MIG | IRS | Mod | Exp |
|---|---|---|---|---|---|
| SAE-12 | **0.411** | **0.238** | 0.508 | **0.930** | 0.890 |
| SAE-6 | 0.294 | 0.052 | 0.479 | 0.928 | 0.877 |
| VLAE-12 | 0.220 | 0.093 | 0.634 | 0.863 | 0.816 |
| $\beta$TCVAE | 0.148 | 0.139 | 0.688 | 0.856 | 0.694 |
| FVAE | 0.095 | 0.044 | 0.664 | 0.916 | 0.722 |
| $\beta$VAE | 0.060 | 0.054 | **0.850** | 0.926 | 0.701 |
| VAE | 0.070 | 0.056 | 0.850 | 0.828 | 0.713 |
| WAE | 0.129 | 0.033 | 0.548 | 0.881 | 0.819 |
| AE | 0.157 | 0.033 | 0.526 | 0.855 | 0.805 |
| AdaAE-12 | 0.159 | 0.022 | 0.481 | 0.893 | **0.905** |

Table 3: Disentanglement and Completeness scores for MPI3D-Sim. (for all these metrics higher is better)

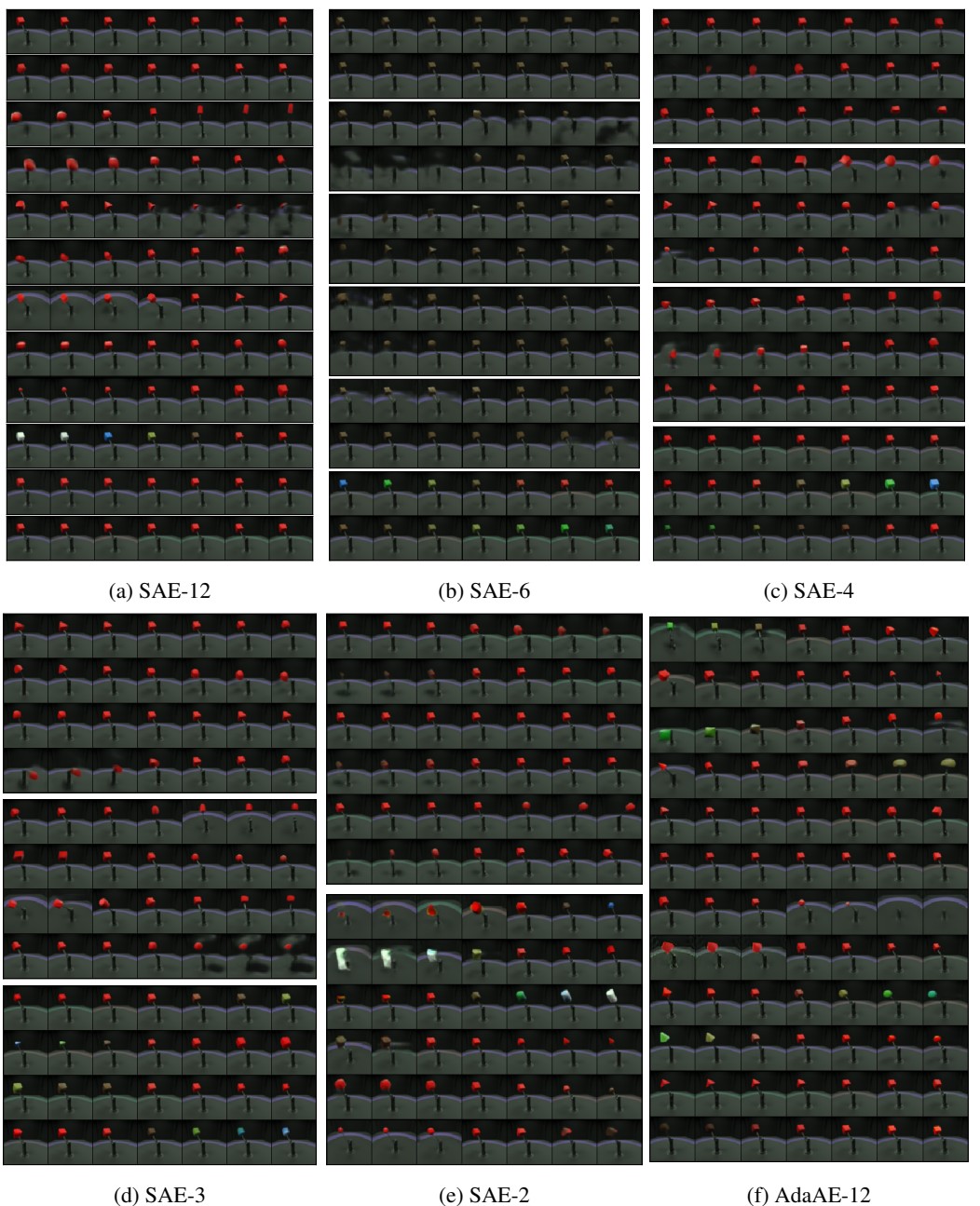

(a) SAE-12          (b) SAE-6          (c) SAE-4

(d) SAE-3          (e) SAE-2          (f) AdaAE-12

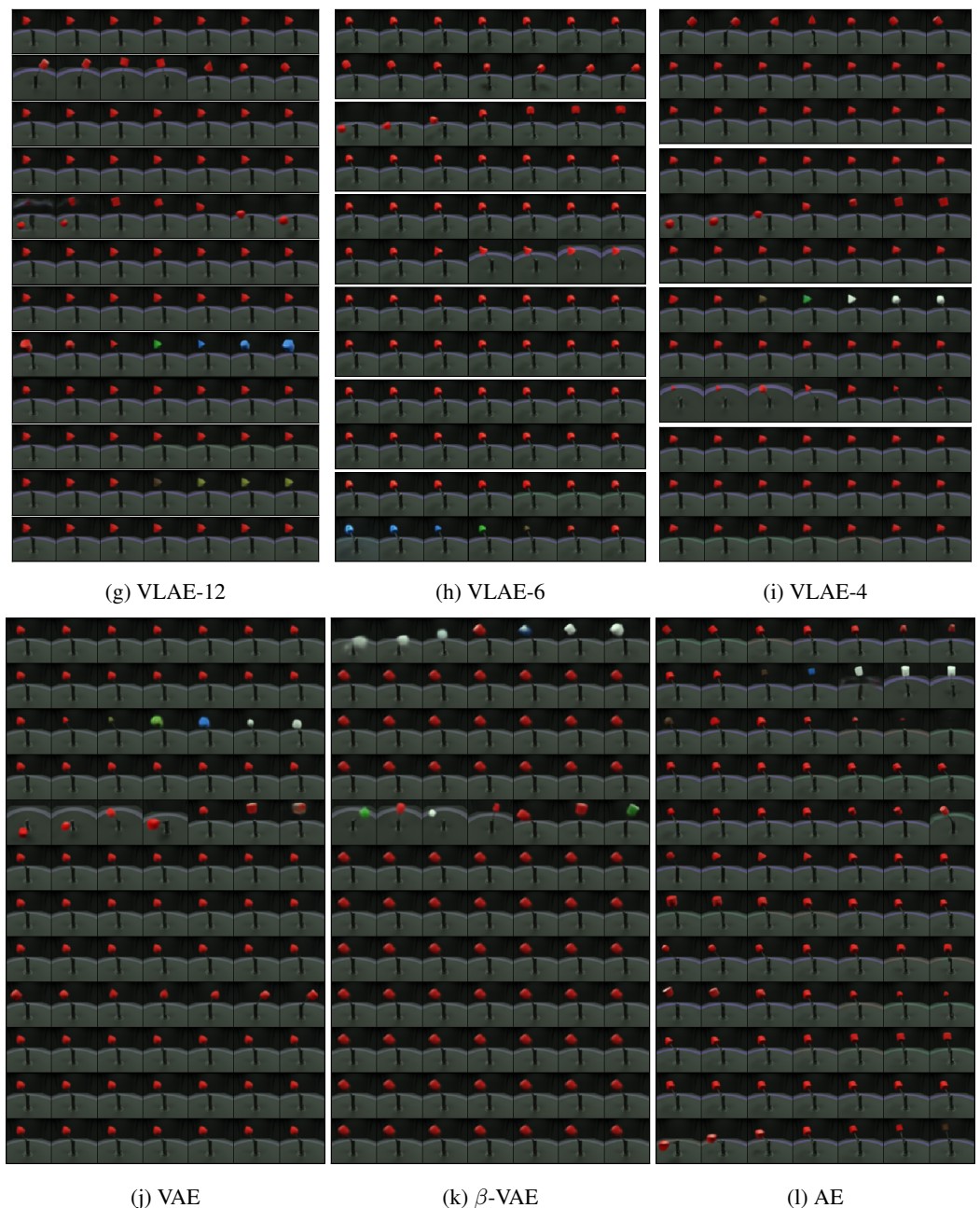

Figure 14: Latent Traversals of several models for MPI3D-Sim. Each row shows the generated image when varying the corresponding latent dimension while fixing the rest of the latent vector. For the SAE and VLAE models, the groups of dimensions that are fed into the same Str-Tfm layer (or ladder rung) are grouped together. Note the disentangled segments achieved by the SAE models and the consistent ordering of factors of variation.

### A.3.6 MPI3D-REAL

| Model | DCI | MIG | IRS | Mod | Exp |
|-------|-----|-----|-----|-----|-----|
| SAE-12 | **0.374** | 0.148 | 0.564 | 0.928 | 0.869 |
| SAE-6 | 0.295 | 0.074 | 0.535 | 0.879 | 0.840 |
| VLAE-12 | 0.291 | **0.217** | 0.579 | 0.914 | 0.805 |
| $\beta$TCVAE | 0.185 | 0.100 | 0.595 | 0.870 | 0.699 |
| FVAE | 0.095 | 0.028 | 0.522 | 0.904 | 0.729 |
| $\beta$VAE | 0.090 | 0.021 | **0.659** | 0.869 | 0.680 |
| VAE | 0.080 | 0.020 | 0.602 | 0.875 | 0.694 |
| WAE | 0.159 | 0.042 | 0.587 | 0.837 | 0.831 |
| AE | 0.143 | 0.048 | 0.563 | 0.858 | 0.804 |
| AdaAE-12 | 0.164 | 0.034 | 0.530 | **0.952** | **0.895** |

Table 4: Disentanglement and Completeness scores for MPI3D-Real. (for all these metrics higher is better)

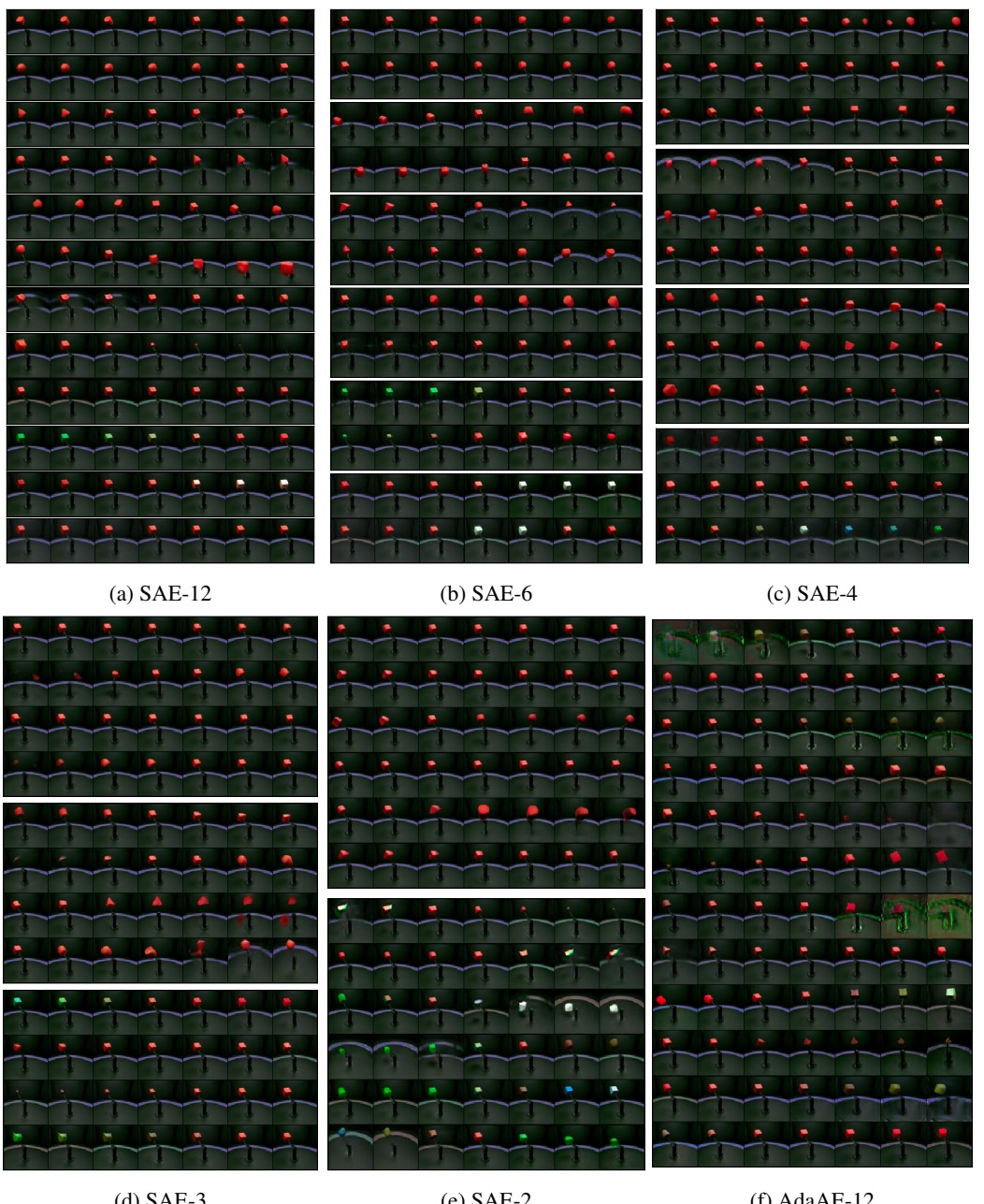

(a) SAE-12                     (b) SAE-6                     (c) SAE-4

(d) SAE-3                     (e) SAE-2                     (f) AdaAE-12

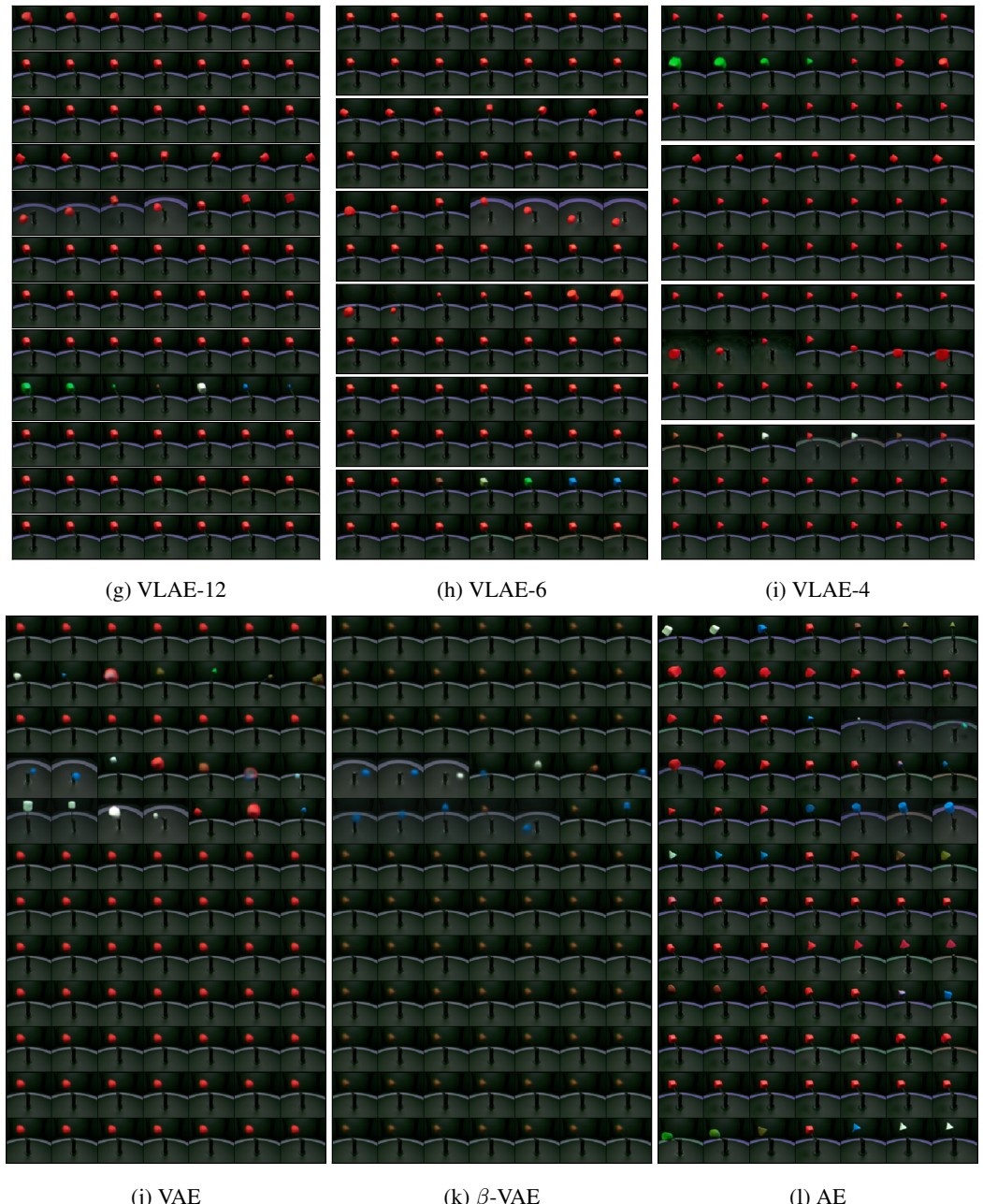

(g) VLAE-12          (h) VLAE-6          (i) VLAE-4

(j) VAE          (k) $\beta$-VAE          (l) AE

Figure 15: Latent Traversals of several models for MPI3D-Real. Each row shows the generated image when varying the corresponding latent dimension while fixing the rest of the latent vector. For the SAE and VLAE models, the groups of dimensions that are fed into the same Str-Tfm layer (or ladder rung) are grouped together. Note the disentangled segments achieved by the SAE models and the consistent ordering of factors of variation.

### A.3.7 CELEB-A

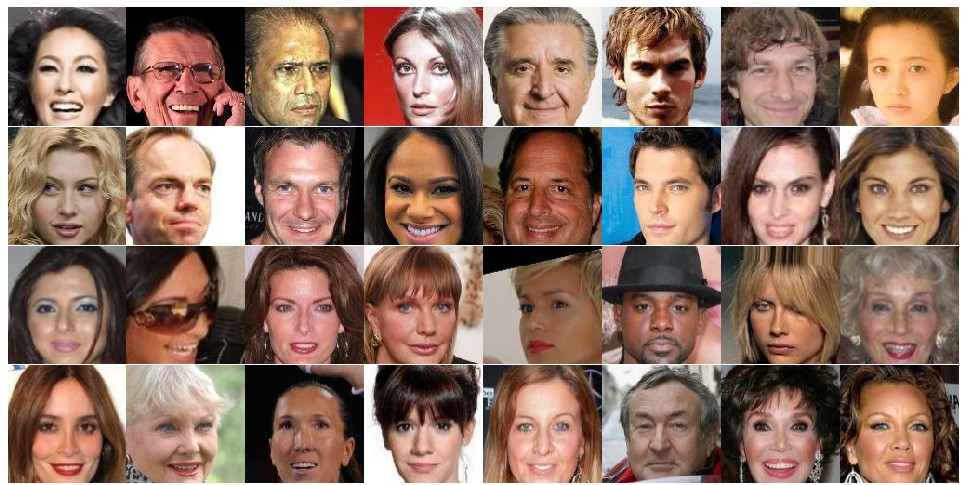

(a) Original samples (from test set)

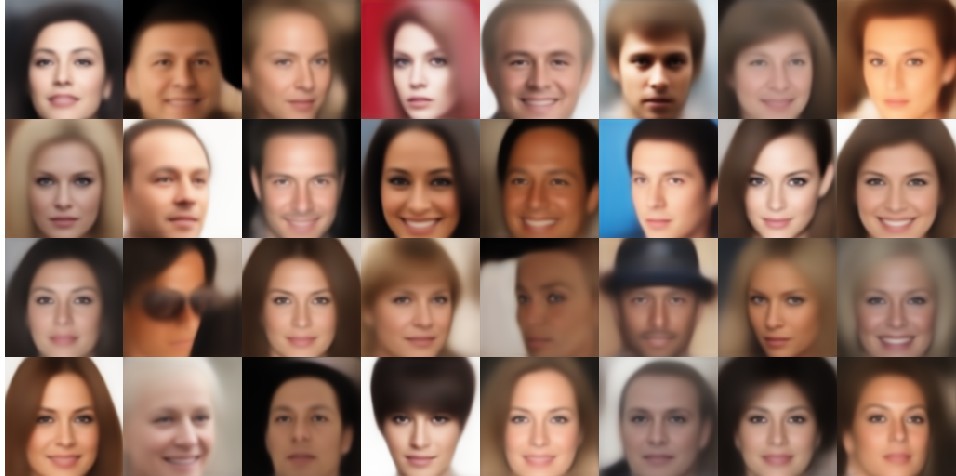

(b) SAE-16 Reconstructions

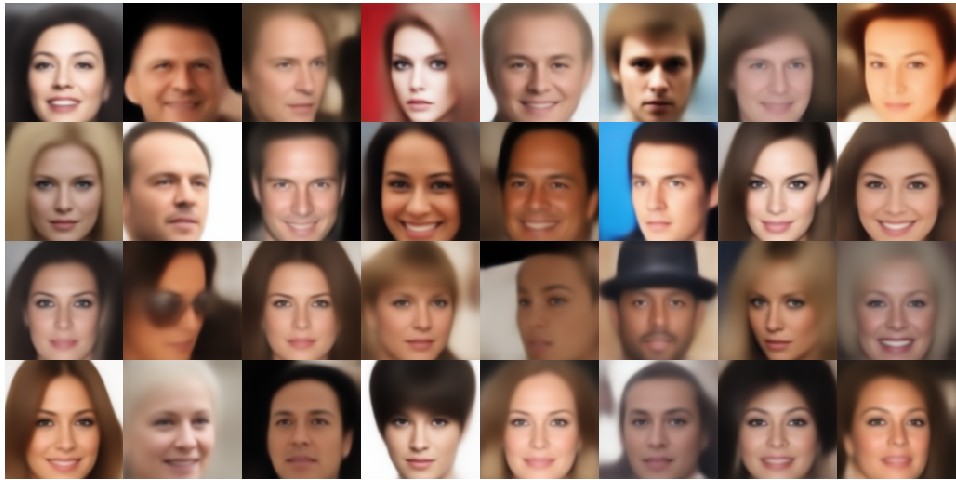

(c) AdaAE-16 Reconstructions

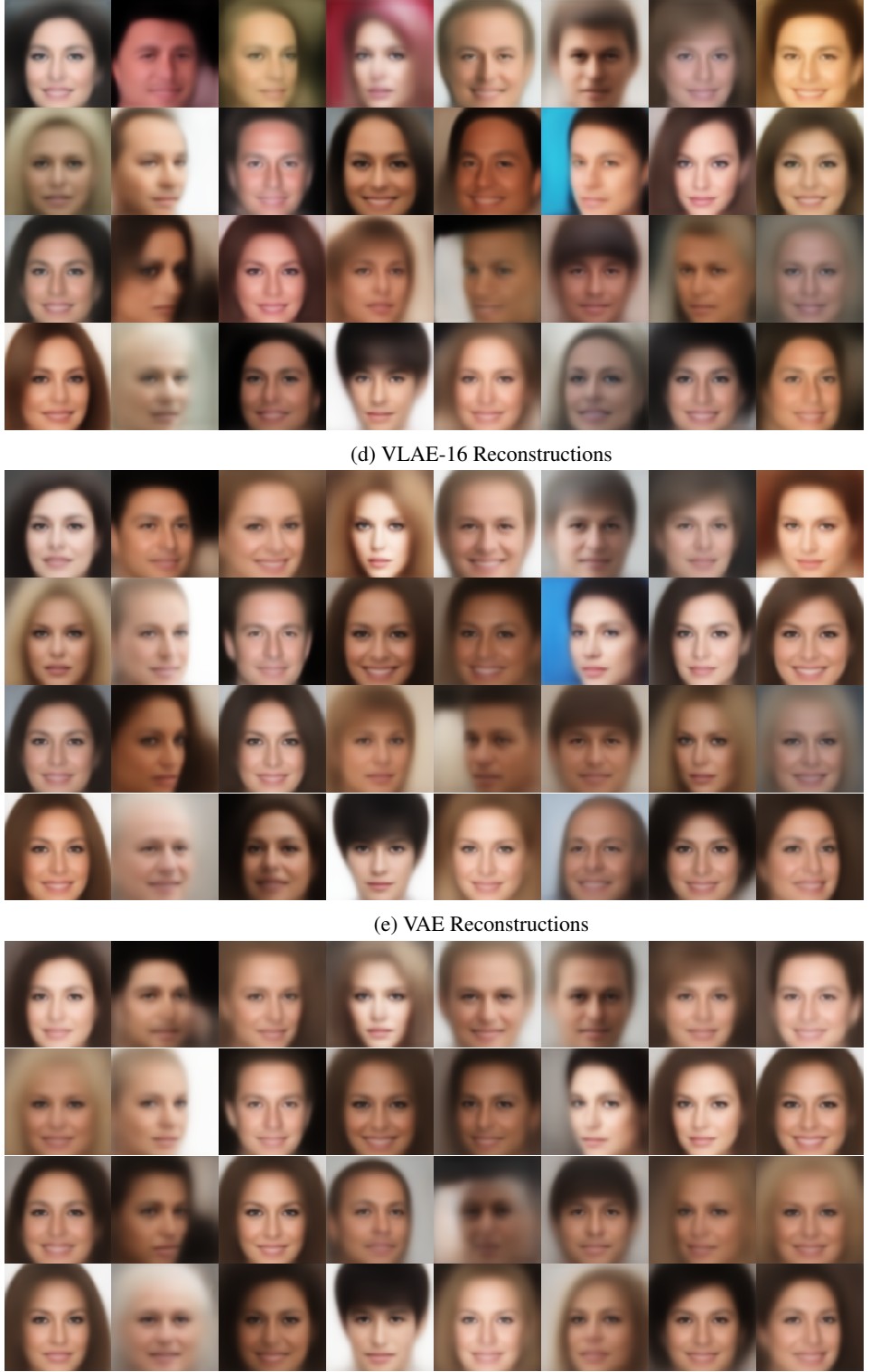

(d) VLAE-16 Reconstructions

(e) VAE Reconstructions

(f) AE Reconstructions

Figure 16: Reconstructed samples using several models trained on CelebA. Note the blurriness of the VAE-based baselines compared to SAE and AdaAE.

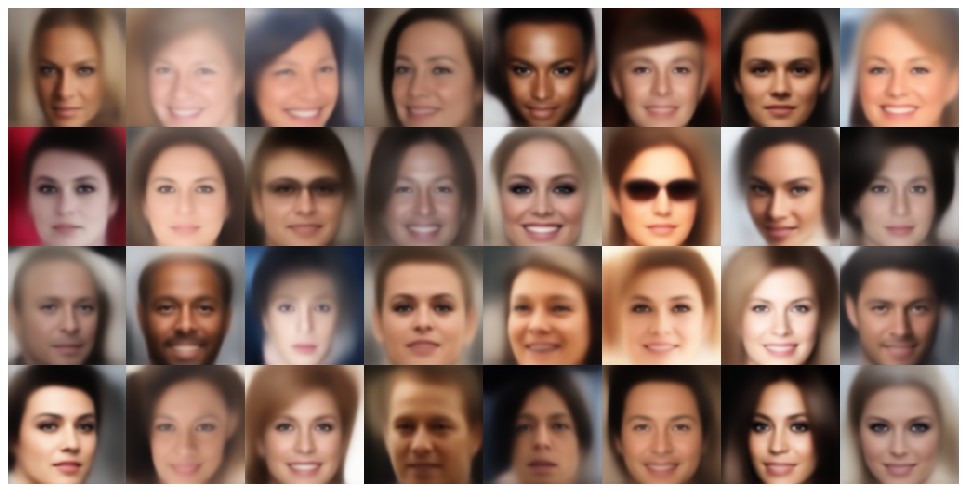

(a) SAE-16 Hybrid Sampling

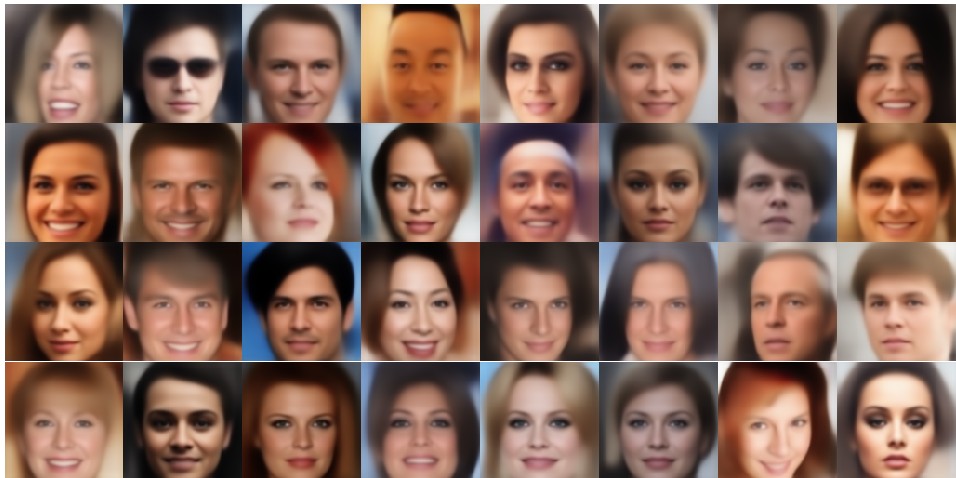

(b) AdaAE-16 Hybrid Sampling

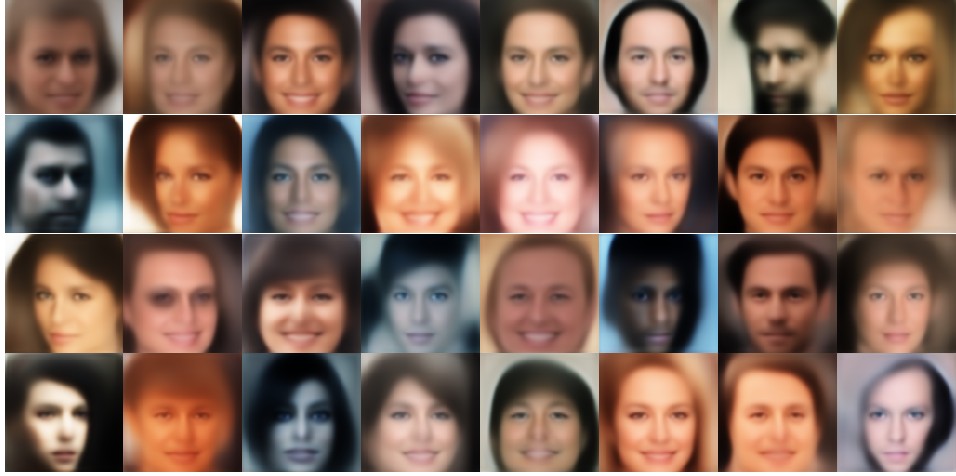

(c) VLAE-16 Hybrid Sampling

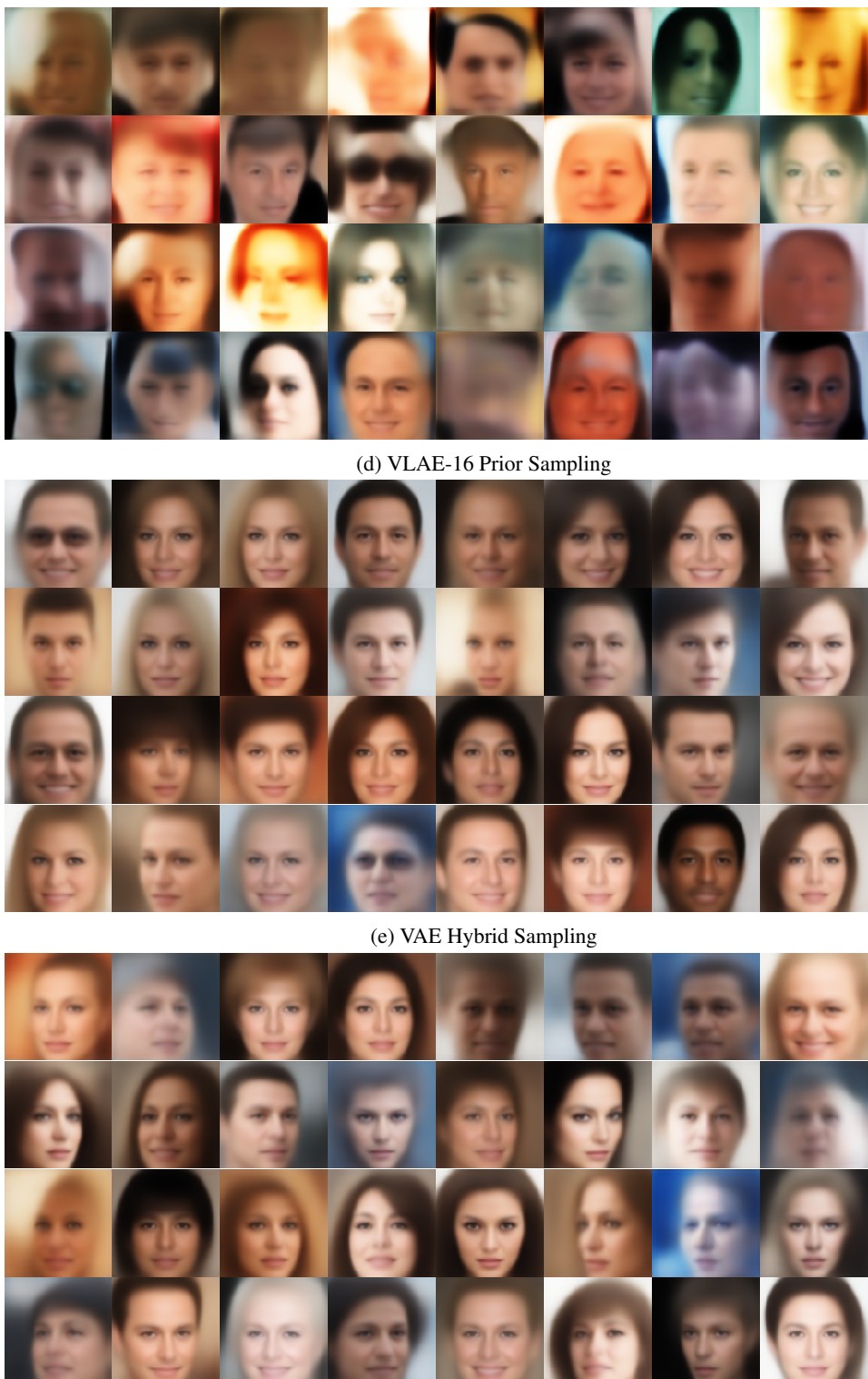

(d) VLAE-16 Prior Sampling

(e) VAE Hybrid Sampling

(f) VAE Prior Sampling

Figure 17: Samples generated using hybrid and prior-based sampling using several models trained on CelebA. Note that the hybrid sampling tends to produce relatively high quality samples both for our proposed SAE and AdaAE architectures as well as baselines.

### A.3.8 RFD

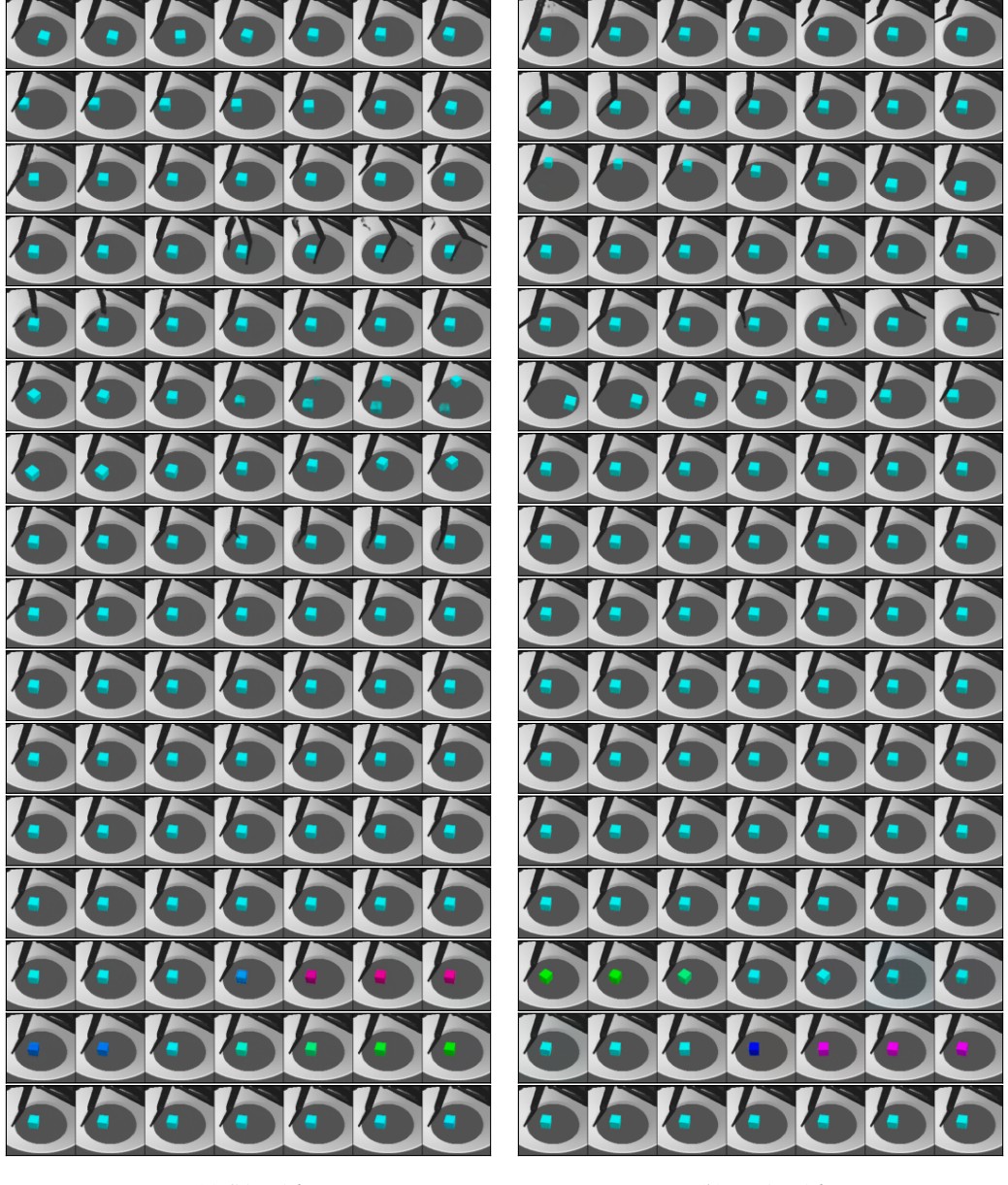

(a) SAE-16                                                     (b) VLAE-16

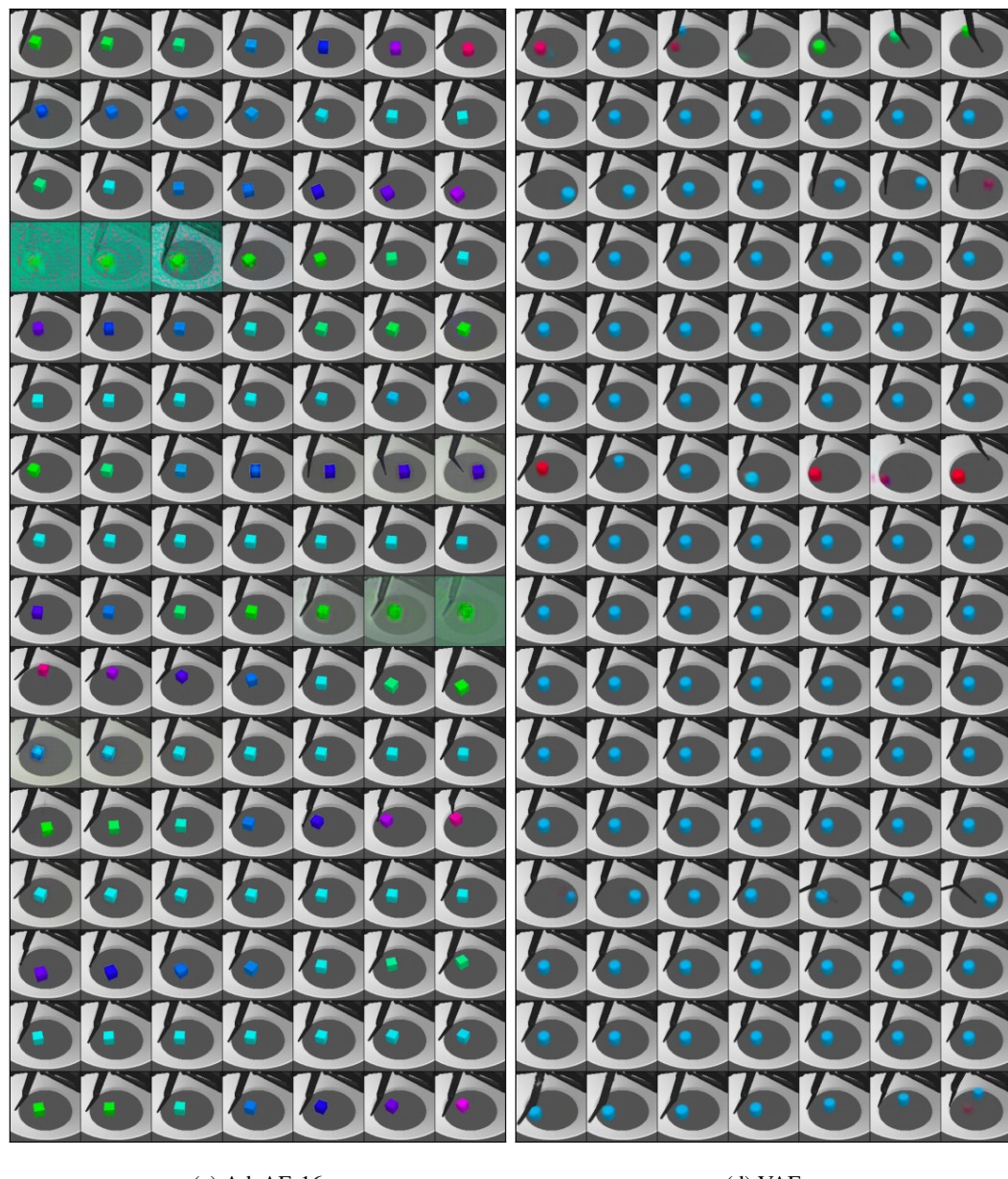

(c) AdaAE-16                                  (d) VAE

Figure 18: Latent traversals for the RFD dataset. Each row corresponds to a 1D traversal of the corresponding latent dimension while the other latent dimensions are fixed. Note the ordering of information in the more structured models like the SAE-16 and VLAE-16.

