# OpenReview forum: "Structure by Architecture: Structured Representations without Regularization"
_ICLR.cc/2023/Conference — ICLR 2023 poster_

### Official Review · Reviewer_ECfH · 2022-10-24

**Confidence:** 4
**Correctness:** 3
**Technical Novelty And Significance:** 3
**Empirical Novelty And Significance:** 3
**Recommendation:** 6

**Clarity, Quality, Novelty And Reproducibility:**

The paper is generally written well. There are sufficient implementation details for reproducibility purposes. The proposed model is related to others existing in the related literature but, to the best of my knowledge, the exact setup is novel.

**Strength And Weaknesses:**

**Strengths**

- The proposed structured decoder which works with a hierarchy of latent variables affecting reconstruction at different scales through StrTfm layers and without regularizing the latent variables to have a pre-set prior distribution is novel.

- The proposed structured decoder is compared to various baselines and shows favorable performance in terms of reconstruction quality, FID, disentanglement, and hierarchy of information encoded in the latent representations.

- The authors provide implementation of their model in the supplementary materials (although I didn’t have a chance to check it closely.)

**Room for improvement**

Please find below clarification questions regarding some of the statements and methodology in the paper.

- *Statistical significance*: Are confidence intervals available for the results in Figure 2 (comparing reconstruction quality for the different models)?

- *Fairness of comparison*: Is it accurate to say that the SAE model has more parameters than the VAE models it is compared to due to the presence of MLP modules in StrTfm layers? If so, could the favorable performance of SAE models in terms of reconstruction quality and FID be partly due to higher capacity of the structured decoder?

- *Initialization of the affine transformation*: Can the authors please elaborate on how the “affine transform for each Str-Tfm layer is initialized to identity” if the hidden layers (I assume the MLPs) are “independently initialized”? I am probably missing some details but are not $\alpha_i$ and $\beta_i$ computed as the outputs of passing $U_i$ through MLP$_i$?

- *Latent traversal*: For the visualizations in Figure 4, when traversing values for one latent dimension, what are the remaining latent dimensions set to? Can the authors elaborate on why there does not seem to be a lot of variation in the generated samples in the top two rows of Figure 4(a)?

- *Information bottleneck acts as regularization*: The latent representations have much lower dimensionality than the inputs ($d\ll  D$) by design which introduces an informational bottleneck that acts as regularization to avoid trivial solutions. How do authors expect the SAE’s performance will be affected as $d$ increases in terms of the reported metrics?


**Summary Of The Paper:**

This paper proposes an autoencoder architecture for structured representation learning. In particular, the proposed model incorporates latent variables at different layers of the convolutional decoder through Structural Transform (StrTfm) layers (inspired by FiLM (Perez et al., 2018) and Ada-In (Karras et al., 2019) layers). In essence, an StrTfm layer applies an affine transformation to the feature maps from the preceding convolutional layer. The latent variable associated with an StrTfm layer determines the magnitude and bias of the affine transformation corresponding to the layer. The latent variables are not regularized by a prior distribution and can be sampled independently for generation based on their training distribution. The proposed model is trained on several datasets and performs favorably when compared to various baselines and benchmarks (different versions of VAEs) based on several evaluation metrics. The evaluation metrics include reconstruction quality (measured by reconstruction loss), visual quality of generated samples (measured by FID), and disentanglement of latent features (measured by DCI, MIG, IRS, Mod/Exp). Additional qualitative analysis shows that the latent space of the structured decoder exhibits some hierarchy where latent codes incorporated at the bottom layers of the decoder (farthest from reconstruction) represent high-level information (such as shape) while latent codes at the top layers of the decoder represent more lower-level information (such as color).

**Summary Of The Review:**

This paper proposes a way to learn a hierarchy of latent representations for generation with an autoencoder. The latent variables are not regularized by a fixed prior distribution and can be sampled independently to generate new samples with relatively high visual fidelity when compared to other models. I have some clarifying questions regarding the methodology and assumptions in the paper and tentatively propose a weak accept.

---

> ### Author Response · Authors · 2022-11-14
> **Thank you for your detailed review and insightful questions!**
>
> - Statistical Significance: For 3D-Shapes, we have results on 5 random seeds in figures 8, 9, and 10. Figure 8 also compares evaluating the reconstruction quality using FID and the pixel-wise binary cross entropy.
> - Fairness of the comparison: This is an excellent point (and we clarified this in appendix A.2.1). To make the comparison between different methods as fair as possible we use the same CNN backbone for all methods. However, as you point out, the SAEs, AdaAEs, and VLAEs all include some additional architectural components which lead to a slightly larger model capacity. However, notably, the VLAEs actually have the largest model capacity (around 5% more than the SAEs) due to the mirrored structure between the encoder and decoder. Consequently, any effect on the performance due to increased capacity is evidently negligible compared to changing the training objective and architecture.
> - Initialization of the affine transformation: Each Str-Tfm layer contains a 3-layer MLP where the hidden layers are initialized using the standard Kaiming uniform sampling and the output layer is initialized such that $\alpha_i \approx 1$ and $\beta_i \approx 0$ by setting the biases of the output layer to 0 and interpreting the MLP output as logits for $\alpha$ (just as is commonly done for the standard deviation of the posterior in VAEs). Consequently, when $U_i$ is passed through $MLP_i$ of the Str-Tfm layer to get $\alpha_i$ and $\beta_i$, the transform is near identity for all Str-Tfm layers even though the hidden layers are all independently initialized. Although all these details are relatively standard (e.g. largely inspired by VAEs and the reparameterization trick), we included this detail in the appendix.
> - Latent traversal: For all the latent traversals (and figure 6a), we start with a latent vector selected from the test set. Then in the latent traversals, one latent variable at a time is varied from $-2\sigma$ to $2\sigma$ where $\sigma$ is the population standard deviation of the marginal estimated from the training set latent vectors. As you point out, the latent traversals of the SAEs show that some of the latent variables are unused (negligible change in the image). As this is a particularly intriguing phenomenon that resembles posterior collapse in VAE-based models, we have added a discussion in appendix A.3.1. In short, since the representation has 12 latent variables and there are only 6 true factors of variation, the independence between latent variables induced by the architecture results in a more compact representation where the excess latent variables are unused.
> - Information bottleneck: This addresses perhaps the most notable limitation of SAEs thus far. Since each pair of latent variables is separated by a convolution block (or, more generally, some nonlinear function), $d$ is proportional to the total depth of the decoder, which may become a limitation for large $d$. Similarly, as $d$ increases it becomes more challenging to learn independent latent variables, thus potentially making hybrid sampling less reliable.
>
>     This effectively boils down to the trade-off between interpretability and performance. Here, we have presented promising results for settings common in the disentanglement community, but for more realistic, challenging datasets, we expect it may be beneficial to combine our proposed inductive biases with a more conventional training setup (similar to Style-GAN) to find a desirable balance between interpretability and performance.

---

> > ### Comment · Reviewer_ECfH · 2022-11-16
> > **Thank you for your detailed response!**
> >
> > Thank you for your detailed response to my questions about the methodology and results! I currently do not have any additional comments.

---

### Official Review · Reviewer_HNCN · 2022-10-24

**Confidence:** 3
**Correctness:** 4
**Technical Novelty And Significance:** 3
**Empirical Novelty And Significance:** 3
**Recommendation:** 8

**Clarity, Quality, Novelty And Reproducibility:**

The paper is clearly written, and the empirical investigation attains a high quality. The technical novelty may not be noteworthy, but the empirical investigation would be valuable for the community. Although I did not try it by myself, the results seem to be reproducible using the attached codes.

**Strength And Weaknesses:**

### Strengths

- Although the method seems simple, it seems to disentangle the representation as nicely reported with intensive experiments.
- Not only the proposed method but also other popular approaches for disentanglement are investigated in different aspects.

### Weaknesses

1. The method's applicability is limited to data for which CNNs are meaningful. This is not a serious weakness, but a brief discussion about the (im)possibility of application to non-image data types would make the paper more complete.

2. Related to the above, the discussion starts by assuming that the data are images without mentioning so at all. This might be natural in the disentanglement community, but for a broader range of audience, limiting the data type first, even just for the sake of discussion, would be helpful.

**Summary Of The Paper:**

An architecture of autoencoders for images that enable disentanglement of the latent variable is proposed. It is contrasted with disentanglement approaches based on regularization. The authors empirically investigate the properties of the proposed method and other disentanglement approaches. One of the conclusions is that the specific architecture of the proposed autoencoder allows the disentanglement of latent representation without any regularization.

**Summary Of The Review:**

This is a solid paper with an intensive empirical study on the disentanglement happening in different types of autoencoders.

---

> ### Author Response · Authors · 2022-11-14
> **Thank you for your positive feedback and comments!**
>
> We are particularly encouraged by your comment that this is a “solid paper with an intensive empirical study”. Since our paper heavily relies on empirical evaluation to identify and characterize useful inductive biases for structured representation learning, we put particular emphasis on providing a wide variety of baselines and datasets as well as including our code and all training details to ensure reproducibility and sufficient evidence for our conclusions.
>
> However, we are curious what makes you think that our proposed method is only applicable to CNNs? While it is true that we only evaluate with image datasets (which we have now clarified in the paper), in principle, neither the structural decoder architecture nor the hybrid sampling method requires a CNN backbone. In fact, the affine transform ($\alpha_i$ and $\beta_i$) of the Str-Tfm layers is the same for all pixels in the intermediate features ($S_{i-1}$). Obviously, this is only speculation as we only used image datasets as those are the most common setting in the disentanglement community, but we would expect the behavior of SAEs on non-image data to be similar.

---

### Official Review · Reviewer_hCAA · 2022-10-25

**Confidence:** 3
**Correctness:** 3
**Technical Novelty And Significance:** 2
**Empirical Novelty And Significance:** 2
**Recommendation:** 6

**Clarity, Quality, Novelty And Reproducibility:**

The paper is quite easy to read and understand, the hybrid sampling method seems quite interesting. The result should be easy to reproduce.

**Strength And Weaknesses:**

Strength
1. The paper showed SAE has better metrics when compared with 6 baselines, and the authors even compared the model performance when hybrid sampling are used in the baseline models.

Weaknesses
1. The hybrid sampling method assumes the independence between the latent variables. The model structure uses one latent variable at a time, but given that all the latent variables are trained at the same time. Not sure how independent are these latent states.
2. Another baseline missing from the paper is autoencoder trained without prior, and sampling based on the proposed hybrid sampling. Because hybrid sampling removed the need of prior distribution for the latent variables, the VAE model can also remove the prior.
3. Base on the generated images Figure4 and Figure 6, the diversity of generated images seems poor.


**Summary Of The Paper:**

This paper proposed SAE, which uses structural decoder that infuses latent information one variable at a time to induce an intuitive ordering of information, and provided a sampling method called hybrid sampling which replies only on independence between latent variables without imposing a prior latent distribution.

The paper compared the metrics with many other baselines, and show improved metrics.

**Summary Of The Review:**

The paper proposed a new model architecture and a new sampling method at the same time, it's difficult to see which gives the real contribution. And it's not clear about the diversity of the generated images.

---

> ### Author Response · Authors · 2022-11-14
> **Thank you for your questions and feedback!**
>
> Unless we have misunderstood you, we have already included the baselines you suggested with an autoencoder without a prior that uses hybrid sampling (called “AE” and “AdaAE”). The AE baseline is a conventional unregularized autoencoder (no prior). Meanwhile, AdaAE is also an unregularized autoencoder but uses a similar architecture to the SAEs except that instead of Str-Tfm layers, FiLM layers are used (thus not splitting up the latent variables and consequently not achieving independence between latent variables). We also already include the results of using hybrid sampling with VAEs (and the other VAE-based baselines), although we have now fixed a minor issue in figure 3 to make that clearer.
>
> We are slightly confused about what you mean by the “diversity of the generated images seems poor” for figures 4 and 6. Figure 4 shows latent traversals - so there we always start with the same latent vector and vary a single variable (corresponding to the row) across each column while keeping all others fixed. Consequently, a disentangled representation is supposed to show low diversity (more precisely, at most one factor of variation should vary across each row). Similarly, for figure 6a, only a quarter of the latent variables (8/32) are varied across each row, so we would expect relatively small changes relative to the diversity of the full dataset. Lastly, 6b is the only figure in the main paper that shows unconditionally generated samples, and there the diversity of samples appears relatively consistent with the real samples (see figure 17 in the appendix for more examples).
>
> The array of baselines is included precisely to understand how our proposed contributions affect the model performance. Comparing the SAEs vs AdaAE (or AE) you can see the effect of the proposed structural decoder architecture alone (since all of those models use hybrid sampling). Meanwhile, comparing the prior-based vs hybrid sampling for any of the VAE based models you can see the effects of hybrid sampling on its own.
>
> We hope this answers your questions and concerns and if so, might also prompt you to increase your recommendation.

---

> > ### Comment · Reviewer_hCAA · 2022-11-25
> > **Thanks for the detailed response.**
> >
> > I currently do not have any additional comments.

---

### Official Review · Reviewer_X8MV · 2022-10-25

**Confidence:** 3
**Correctness:** 2
**Technical Novelty And Significance:** 2
**Empirical Novelty And Significance:** 2
**Recommendation:** 6

**Clarity, Quality, Novelty And Reproducibility:**

Paper is clearly written and references some of the key works in this area. Appendix provides sufficient details to reconstruct the training setup and all of the model parameters which should be sufficient to reproduce the results. Some figures (e.g. Fig 3 is missing a legend for the type of sampling used).


**Strength And Weaknesses:**

Strengths:
* Carries out experiments on several datasets.
* Paper suggests a simple and clearly explained approach

Weaknesses:
* Model is evaluated on rather simple test problems
* Some natural baselines like VQ-VAE are not represented
* Limitations/caveats arising from the choice of hybrid sampling are not clearly addressed/stated/investigated. (e.g. effect on FID score etc.)


**Summary Of The Paper:**

The paper considers an autoencoder architecture where the structure in the latent space is induced by architecture of the decoder, which uses latent variables sequentially to generate samples from the target distribution. Rather than imposing a prior distribution on the latent space authors use a “hybrid sampling” approach to generate samples that involves picking components of latent vectors from the cross product of a set of randomly chosen examples from the dataset. Authors study the effectiveness of their approach using FID scores, disentanglement scores and a series of visual inspections on several datasets including (MPI3D, Celeb-A, 3D-Shapes). Authors find that their models perform favorably with respect to other baselines, especially on the disentanglement metrics.


**Summary Of The Review:**

While the paper presents a new and interesting approach to adding structure to a latent representation by modifying the decoder architecture and how the latent vectors are converted to samples from the data distributions, I’m not convinced that current results offer evidence to support the claim that this is a principled step in a right direction for representation learning. I might have misunderstood parts of the paper in which case I’d love to be convinced otherwise.

My main concerns are:
(1) IIUC objective scores like log likelihood of the test dataset are not possible to calculate with hybrid type sampling, while scores like FID are easily fooled by memorization.
(2) Other generative models have not been considered (e.g. VQ-VAE-2 [1], RG-Flow[2] etc.)




[1] https://arxiv.org/pdf/1906.00446.pdf
[2] https://iopscience.iop.org/article/10.1088/2632-2153/ac8393/pdf

---

> ### Author Response · Authors · 2022-11-14
> **Thank you for your review and valuable critique.**
>
> We are encouraged by your comment that our work “presents a new and interesting approach to adding structure to a latent representation” - which is indeed one of our main goals. However, could you expand on why you are nevertheless “not convinced that current results offer evidence to support the claim that this is a principled step in the right direction for representation learning”?
>
> It appears there is a mismatch between our focus and your suggestions which we hope to rectify. In short, our goal is to learn an *interpretable structured representation* without sacrificing sample fidelity rather than addressing unconditional generative modeling alone. In this setting (as is common in the disentanglement community), it is crucial not only to evaluate the quality of the generated samples but also to understand the representation (e.g. using disentanglement metrics) - which is also why synthetic datasets are commonly used to enable quantitative analysis.
>
> Consequently, two of the references you mentioned (VQ-VAE and VQ-VAE-2) are not appropriate comparisons because, firstly, the latent spaces orders of magnitude larger (~5000 DOFs for VQ-VAE-2, compared to 32 in our largest models) making evaluating disentanglement impractical. Furthermore, for VQ-VAE, the resulting latent space is uninterpretable, even necessitating the prior to be learned post hoc with an autoregressive model, thus making a comparison to methods impossible on anything other than sample fidelity alone.
>
> The other reference, RG-Flow, also has significant fundamental disparities to our body of work making comparison impractical. The use of normalizing flows necessitates the latent space of RG-Flow to be equal to the images, thus making any performance difficult to attribute to either the much larger latent space or differing inductive biases. Additionally, with RG-Flow, any semantically meaningful features must be found manually by trial and error, while the vast majority of latent space are uninterpretable (also resulting in issues with the disentanglement metrics). In contrast, the more intuitive structure of SAEs order the latent information by how linear a feature is w.r.t. the pixel space.
>
> Regarding your first concern, we agree that there are issues with the evaluation metrics for generative models such as FID - however, this is still a largely open problem, and given its popularity and that FID allows direct comparison with any other generative method (including GANs), we believe FID is satisfactory for our purposes. We also include generated samples of all our models in the appendix and find good qualitative agreement with the FID scores, and we compare the quantitative results using FID and pixel-wise metrics for reconstruction in figure 8.
>
> Lastly, thank you for pointing out that, for figure 3, the key specifying the sampling method was not always showing up correctly. It appears to be a problem with our rendering of the figures, so we have temporarily replaced the PDF figures with PNGs.

---

> > ### Comment · Reviewer_X8MV · 2022-11-28
> > **Reply**
> >
> > Thank you for your reply and clarifications.
> >
> > I agree with the authors that models with much higher dimensional latent space are harder to interpret. After reviewing disentanglement score more closely I'm leaning to side with other reviewers that this is an interesting result and hence rising my score to 6.
> >
> > As for the RG-Flow, while I agree that with all normalizing flow models the total latent space has the same size as inputs, a hierarchical structure would explicitly select a small number of latent dimensions that carry semantic information. Wouldn't selecting `n` top level features, combined with sampling of the other latent values from the prior distribution provide a way to compare such different approaches on the disentanglement scores?

---

> > > ### Author Response · Authors · 2022-12-13
> > > **Note on RG-Flow**
> > >
> > > While we agree in principle that the structure of RG-Flow lends itself to more careful analysis of disentanglement, the authors of RG-Flow do not perform any such quantitative evaluation, making it difficult to tell how RG-Flow compares to more conventional methods. We suspect there are several reasons: (1) as the hierarchical structure of the variables is based on receptive fields (rather than model capacity in SAEs), the disentanglement may only by possible for factors of variation that consistently depend on the same regions of the image. (2) Although the hierarchical model structure specifies the high-level variables as you say, for evaluation it is unclear how many variables should be used to compute the disentanglement scores because the more variables you include the more true factors will be captured but then you also start using variables quite far down the hierarchy which are probably highly entangled.
> > >
> > > Thank you for raising your score!

---

### Author Response · Authors · 2022-11-14
**Thank you for the discussion!**

Overall, we are encouraged by the largely thoughtful responses of the reviewers, all of which expressed an interest in our project, and that the paper was clearly written and easy to understand.

Based on all the reviewers’ helpful feedback, we have made numerous minor changes to the paper including adding several clarifying sections in the appendix. We noticed that several reviewers raised concerns that, to us, appear to have already been covered in the paper, but we are happy to further clarify anything missing.

We hope this revision will further help the reviewers reach a consensus on the value of our work and foster further discussion.

---

### Comment · Area_Chair_2XZv · 2022-11-15
**Please respond to revised version**

Dear reviewers,

Your response to the revised version of the paper would be highly appreciated.

Kind regards,
Your AC

---

### Decision · Program_Chairs · 2023-01-20

**Decision:**

Accept: poster

**Justification For Why Not Higher Score:**

The reviews agree that the method is interesting, and although simple, seems to be effective. The empirical experimentation is interesting and sufficient, and show promising results in terms of FID and disentanglement.

**Justification For Why Not Lower Score:**

A weakness is that the experiments are not done on a very large scale, such that the performance of the method at scale is still unclear.

**Metareview: Summary, Strengths And Weaknesses:**

Reviewer X8MV: Rating 6, Conf 3
Reviewer hCAA: Rating 6, Conf 3
Reviewer HNCN: Rating 8, Conf 3
Reviewer ECfH: Rating 6, Conf 4

This paper proposes a way to learn a generative model through a hierarchy of latent representations for generation with an autoencoder. The latent variables are not regularized by a fixed prior distribution and can be sampled independently to generate new samples with relatively high visual fidelity when compared to other models.

The reviews agree that the method is interesting, and although simple, seems to be effective. The empirical experimentation is interesting and sufficient, and show promising results in terms of FID and disentanglement. A weakness is that the experiments are not done on a very large scale, such that the performance of the method at scale is still unclear.

After discussions, Reviewer X8MV upgraded his rating to a 6. Review hCAA is unfortunately not a very high-quality review. The author tried to address his concerns, to which the reviewer did not really respond, making me doubt whether he read them. Reviewer ECfH is positive as well, and raised some questions, which were answered in detail by the authors.

Based on reviewer feedback, the authors made updates to the paper including adding several clarifying sections in the appendix.

**Note From Pc:**

if the above contains the word "oral" or "spotlight" please see: "oral" presentation means -> notable-top-5% and "spotlight" means -> notable-top-25%. As stated in our emails, we are disassociating presentation type from AC recommendations